biomechanics, physiology

force–length and force–velocity relationship, enthalpy–velocity relationship, triceps surae, endurance running, strength training, tendon stiffness

**Author for correspondence:**
Sebastian Bohm
e-mail: sebastian.bohm@hu-berlin.de

# Enthalpy efficiency of the soleus muscle contributes to improvements in running economy

Sebastian Bohm[1,2], Falk Mersmann[1,2], Alessandro Santuz[1,2] and Adamantios Arampatzis[1,2]

[1]Department of Training and Movement Sciences, Humboldt-Universität zu Berlin, Philippstr. 13, 10115 Berlin, Germany
[2]Berlin School of Movement Science, Humboldt-Universität zu Berlin, Berlin, Germany

SB, 0000-0002-5720-3672; FM, 0000-0001-7180-7109; AS, 0000-0002-6577-5101; AA, 0000-0002-4985-0335

During human running, the soleus, as the main plantar flexor muscle, generates the majority of the mechanical work through active shortening. The fraction of chemical energy that is converted into muscular work (enthalpy efficiency) depends on the muscle shortening velocity. Here, we investigated the soleus muscle fascicle behaviour during running with respect to the enthalpy efficiency as a mechanism that could contribute to improvements in running economy after exercise-induced increases of plantar flexor strength and Achilles tendon (AT) stiffness. Using a controlled longitudinal study design ($n = 23$) featuring a specific 14-week muscle–tendon training, increases in muscle strength (10%) and tendon stiffness (31%) and reduced metabolic cost of running (4%) were found only in the intervention group ($n = 13$, $p < 0.05$). Following training, the soleus fascicles operated at higher enthalpy efficiency during the phase of muscle–tendon unit (MTU) lengthening (15%) and in average over stance (7%, $p < 0.05$). Thus, improvements in energetic cost following increases in plantar flexor strength and AT stiffness seem attributed to increased enthalpy efficiency of the operating soleus muscle. The results further imply that the soleus energy production in the first part of stance, when the MTU is lengthening, may be crucial for the overall metabolic energy cost of running.

## 1. Introduction

Habitual bipedalism has been recognized as a defining feature of humans [1], and an exceptional endurance running ability has been linked to the evolution of the human lineage [2]. Economy, which is the mass-specific rate of oxygen uptake or metabolic energy consumption at a given speed [3,4], plays a crucial role in endurance running performance [5]. The cost of generating force and work through muscles to support and accelerate the body mass is the main source of metabolic energy expenditure during locomotion [6]. The force–length–velocity potential of muscles (defined as the fraction of maximum force according to the force–length [7] and force–velocity relationships [8]) at which muscles operate during running [9,10] largely dictates the required active muscle volume and consequently the energetic cost of contraction [3,9,11].

In human running, the triceps surae is the major contributor to propulsion and the main plantar flexor muscle group that transmits force through the Achilles tendon (AT) [12], consuming a significant amount of metabolic energy [13]. In earlier studies, we provided evidence that both the contractile capacities of the triceps surae and the mechanical properties of the AT (i.e. its stiffness) influence running economy [14,15]. We found that the most

economical runners feature a combination of higher plantar flexor muscle strength and AT stiffness [14], and that a specific training of muscle strength and AT stiffness can, in fact, improve running economy [15]. Although the association of AT stiffness and energetic cost of running has been confirmed by other research groups [16,17], the underlying physiological mechanisms remain unclear.

The soleus is the greatest muscle of the triceps surae [18] and generates the majority of work/energy to lift and accelerate the body [12] by actively shortening throughout the entire stance phase of running [9,19]. In the first part of the stance phase, the fascicle shortening is paralleled by a lengthening of the muscle–tendon unit (MTU) [9], indicating that a part of the body's mechanical energy is stored as strain energy in the AT, but also that the fascicles generate work and save this work as strain energy in the AT. In the second part of the stance phase, where the MTU shortens (propulsion phase), the tendon strain energy is returned to the body and contributes to the ongoing work generation [9]. The metabolic cost of generating work by active shortening of muscles depends on the velocity of the shortening [20]. The enthalpy efficiency (or mechanical efficiency) quantifies the fraction of chemical energy from ATP hydrolysis that is converted into mechanical muscular work [21]. The relation of enthalpy efficiency and shortening velocity shows a steep increase at low velocities with the peak at around 20% of the maximum shortening velocity [21,22]. During submaximal running, the soleus operates below the optimal velocity for maximal efficiency [9], suggesting that small changes in the shortening velocity may substantially influence the enthalpy efficiency of the soleus muscular work production.

The mechanical interaction of the soleus muscle with the series AT regulates the fascicle shortening dynamics. The AT takes over a great part of the length changes of the entire soleus MTU, thereby decoupling the muscle fascicle and MTU behaviour and, beside the storage and release of strain energy, allowing the fascicles to operate at velocities favourable for economical force generation [9,19]. The mechanical properties of the tendon in combination with the strength capacity of the muscle may determine the amount of fascicle decoupling during the stance phase of running. However, similar to an increase in muscle strength [23], tendons can adapt to periods of higher mechanical loading by increasing stiffness [24]. Our earlier findings of improved energetic cost after an exercise-induced increase in AT stiffness and plantar flexor muscle strength evidenced a direct association between a balanced adaptation of tendon and muscle and improvements in running economy [15]. Considering a given work produced by the soleus muscle during the stance phase, the energetic cost depends on the enthalpy efficiency under which this muscular work is generated. Assuming that a combination of increased plantar flexor strength and AT stiffness may influence the soleus fascicle shortening pattern, the overall enthalpy efficiency might improve. This would provide an explaining mechanism to the previously reported improvements in running economy following effective muscle–tendon training [15]. To the best of our knowledge, no study has experimentally examined the operating soleus muscle fascicles with respect to the enthalpy efficiency and its association to the energetic cost of running.

Here, we investigated the effect of a specific muscle-tendon training, which has been shown to increase plantar flexor strength and AT stiffness [15], on the enthalpy efficiency of the operating soleus fascicles during running. Based on our earlier study [15], we expected an improvement in running economy after 14 weeks of training. We hypothesized that the training-induced increase in plantar flexor muscle strength and AT stiffness modulates the soleus fascicle velocity pattern throughout the stance phase towards velocities associated with a higher enthalpy efficiency, thereby reducing the energetic cost of running.

## 2. Methods

### (a) Participants and experimental design
A statistical power analysis was performed *a priori* and revealed a required sample size of $n = 12$ for the intervention group (see electronic supplementary material for details). Considering potential dropouts, we recruited 36 participants and randomly assigned them to an intervention ($n = 19$) or control group ($n = 17$). Inclusion criteria were an age of 20–40 years, at least two running sessions weekly on a recreational basis and no muscular–tendinous injuries in the previous year. Only habitual rearfoot-striking runners were considered because it is the most common foot strike pattern [25] and also to avoid potential confounding effects of the strike pattern on our outcome measures. To quantify the foot strike pattern, we assessed the strike index [26] (i.e. centre of pressure position with respect to the heel relative to foot length at touchdown) during a pre-test session (0 equals rearfoot-striking, <0.3 inclusion threshold). Twenty-three participants completed the study, of which 13 were the intervention group (age $29 \pm 5$ years, height $178 \pm 8$ cm, body mass $73 \pm 8$ kg, four females) and 10 the control group ($31 \pm 3$ years, $175 \pm 10$ cm, $70 \pm 11$ kg, seven females). For the intervention group, the same 14-week muscle–tendon training was added to the regular ongoing training habits as in our earlier study [15]. Before and after the intervention period, the maximal plantar flexion moment and AT stiffness as well as energetic cost of running at $2.5$ m s$^{-1}$ were assessed in both groups. To explain the expected improvements in energetic cost following the training, we experimentally determined (i) the foot strike pattern, joint kinematics and temporal gait parameters as well as (ii) the soleus MTU and fascicle behaviour in addition to the electromyographic (EMG) activity during running. We further determined (iii) the soleus force–fascicle length relationship and force–fascicle velocity relationship in order to calculate the force–length and force–velocity potential of the fascicles during running (i.e. fraction of maximum force according to the force–length and force–velocity curve [9,10,27]) and assessed (iv) the enthalpy efficiency–fascicle velocity relationship to calculate the efficiency of the soleus muscle during running. Because changes in running economy were not expected without any intervention [15], the assessment of the fascicle behaviour was not conducted in the controls. The university ethics committee approved the study, and participants gave written informed consent in accordance with the Declaration of Helsinki.

### (b) Exercise protocol
The supervised and biofeedback-based resistance training was performed for 14 weeks and was characterized by five sets of four repetitive isometric ankle plantar flexion contractions (3 s loading and 3 s relaxation) at 90% of the maximum voluntary contraction (MVC) strength (adjusted every two weeks), three to four times a week (see electronic supplementary material for illustration). This loading regimen has been shown to provide a sufficient magnitude and duration of tendon strain to promote AT adaptation in addition to increases in plantar flexor muscle strength [15,24,28].

## (c) Strength of the plantar flexors and Achilles tendon stiffness

The plantar flexor strength of the right leg was measured using an inverse dynamics approach. For the determination of AT stiffness, ramp-MVCs were conducted and the force applied to the AT was calculated as quotient of joint moment and individual tendon lever arm, which was determined using the tendon-excursion method. The corresponding AT elongation was analysed based on the displacement of the gastrocnemius medialis-myotendinous junction visualized by ultrasonography. Stiffness was calculated between 50 and 100% of the maximum tendon force and strain by dividing elongation by resting length (see electronic supplementary material for details).

## (d) Energetic cost of running

During an 8 min running trial on a treadmill at 2.5 m s$^{-1}$, expired gas analysis was conducted and the rate of oxygen consumption ($\dot{V}O_2$) and carbon dioxide production ($\dot{V}CO_2$) was calculated as the average of the last 3 min [15]. Running economy was then expressed in units of energy as Energetic cost $= 16.89 \cdot \dot{V}O_2 + 4.84 \cdot \dot{V}CO_2$, where energetic cost is presented in (W kg$^{-1}$) and $\dot{V}O_2$ and $\dot{V}CO_2$ in (ml s$^{-1}$ kg$^{-1}$) [4,29]. The steady state was visually confirmed by the rate of $\dot{V}O_2$ during each trial, and a respiratory exchange ratio (RER) of <1.0 was controlled for during the post analysis (see electronic supplementary material for details).

## (e) Joint kinematics and foot strike pattern

Kinematics of the right leg were captured (250 Hz) by a Vicon motion capture system (Nexus 1.8, Vicon, Oxford, UK) using anatomical-referenced markers [9]. The touchdown and the toe-off were determined from the kinematic data as consecutive minimum in knee joint angle over time [30]. The foot strike pattern was analysed by means of the strike index [26]. A self-developed algorithm [25] was used to calculate the strike index from the plantar pressure distribution (120 Hz) captured by the integrated pressure plate (FDM-THM-S, Zebris Medical GmbH, Isny, Germany).

## (f) Soleus muscle-tendon unit length changes, fascicle behaviour and electromyographic activity during running

During an additional 3 min running trial at the same speed, kinematics of the ankle joint served to calculate the length change of the soleus MTU as the product of ankle angle changes and the previously assessed individual AT lever arm [31]. The initial soleus MTU length was determined at a neutral joint angle using the previously reported regression equation by Hawkins & Hull [32]. Ultrasonic images of the soleus muscle fascicles were obtained synchronously at 146 Hz (Aloka Alpha7, Tokyo, Japan). The probe (6 cm linear array, 13.3 MHz) was mounted over the medial soleus muscle belly. The fascicle length was post-processed from the images using a semi-automatic tracking algorithm [33] (figure 1), and corrections were made if necessary. At least nine steps were averaged [10]. The velocities of MTU and fascicles were calculated as the first derivative of the lengths over the time. Synchronized surface EMG of soleus was measured (1000 Hz) by means of a wireless EMG system (Myon m320RX, Baar, Switzerland) and is presented as normalized to the maximum EMG value observed from the individual MVCs [9].

## (g) Soleus force–length, force–velocity and efficiency–velocity relationship

To determine the soleus force–fascicle length relationship (for details [9]), the participants were placed in the prone position on

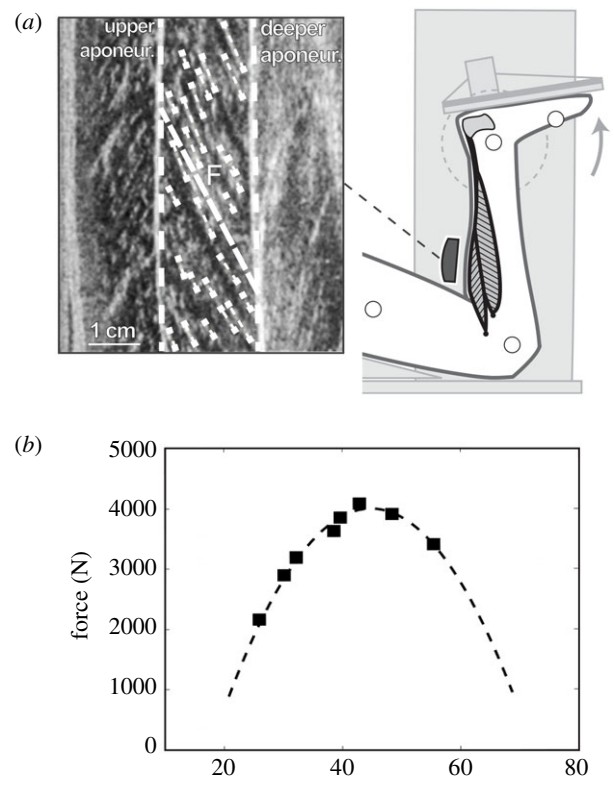

**Figure 1.** (a) Experimental set-up for the determination of the soleus force–fascicle length relationship. MVCs at eight different joint angles were performed on a dynamometer. During the MVCs, the soleus muscle fascicle length was measured by ultrasonography as an average ($F$) of multiple fascicle portions (short-dashed white lines) identified from the images. (b) Exemplary force–length relationship of the soleus fascicles obtained from the MVCs (squares) and the respective second-order polynomial fit (dashed line).

the bench of the dynamometer (Biodex Medical, Shirley, NY) with the knee fixed in a flexed position (figure 1) to restrict the contribution of the bi-articular gastrocnemius muscle to the plantar flexion moment (approx. 120°) [34]. MVCs were performed with the right leg in eight different joint angles, and the joint moments and force acting on the AT were calculated as described in section 2c above. The corresponding soleus fascicle behaviour was captured synchronously at 30 Hz by ultrasonography, and fascicle length was measured accordingly (figure 1). The probe remained attached between the running trial and MVCs. An individual force–fascicle length relationship was calculated by means of a second-order polynomial fit (figure 1), giving the maximum force ($F_{max}$) and optimal fascicle length for force generation ($L_0$).

The force–velocity relationship of the soleus was assessed using the classical Hill equation [8] and the maximum fascicle shortening velocity ($V_{max}$) and constants of $a_{rel}$ and $b_{rel}$. For $V_{max}$, we took reported values of human soleus type 1 and 2 fibres [35], adjusted those for physiological temperature [36] and applied an average fibre-type distribution (81% type 1 fibres and 19% type 2 [9]), giving $V_{max}$ as 6.77 $L_0$ s$^{-1}$ [9]. $a_{rel}$ was calculated as $0.1 + 0.4 \times$ type 2 fibre percentage [37], which equals to 0.175. The product of $a_{rel}$ and $V_{max}$ gives $b_{rel}$ as 1.182 [37]. Based on the assessed force–length and force–velocity relationships, it was possible to calculate the individual force–length and force–velocity potential of soleus as a function of the fascicle length (figure 1) and velocity during running [9,10,27].

Furthermore, we determined the enthalpy efficiency–shortening velocity relationship for the soleus fascicles to calculate the enthalpy efficiency of the soleus as a function of the fascicle velocity during running. We referred to the

**Table 1.** Maximal plantar flexion moment and Achilles tendon stiffness as well as energetic cost, foot strike index and temporal step characteristics during running before and after the training period for the intervention and control group (mean ± s.d., effect size $g$).

| | intervention (n = 13) | | | control (n = 10) | | |
|---|---|---|---|---|---|---|
| | pre | post | $g$ | pre | post | $g$ |
| moment (Nm kg$^{-1}$)[a] | 3.12 ± 0.48 | 3.44 ± 0.37[c] | 0.77 | 3.10 ± 0.46 | 2.99 ± 0.32 | 0.32 |
| stiffness (kN strain$^{-1}$)[a] | 85 ± 36 | 111 ± 59[c] | 0.67 | 73 ± 29 | 71 ± 28 | 0.10 |
| energy cost (W kg$^{-1}$)[b] | 10.6 ± 0.6 | 10.2 ± 0.7[c] | 0.74 | 11.2 ± 1.0 | 11.1 ± 1.0 | 0.12 |
| strike index | 0.08 ± 0.12 | 0.10 ± 0.16 | 0.09 | 0.06 ± 0.03 | 0.06 ± 0.03 | 0.05 |
| stance time (ms) | 310 ± 23 | 316 ± 23 | 0.29 | 327 ± 17 | 324 ± 23 | 0.34 |
| flight time (ms) | 53 ± 31 | 53 ± 24 | 0.01 | 50 ± 31 | 54 ± 31 | 0.48 |
| cadence (steps min$^{-1}$) | 160 ± 11 | 159 ± 9 | 0.39 | 162 ± 9 | 161 ± 9 | 0.26 |

[a]Significant time by group interaction effect ($p < 0.05$).
[b]Significant main effect of time ($p < 0.05$).
[c]Significant difference (*post hoc* analysis) to pre ($p < 0.05$).

**Table 2.** Ankle and knee joint angles at touchdown, toe-off and at the maximal ankle dorsiflexion and knee flexion angle, respectively, during running before and after the training intervention (mean ± s.d., effect size $g$, $n = 13$).

| | touchdown | | | toe-off | | | maximum dorsiflexion/knee flexion | | |
|---|---|---|---|---|---|---|---|---|---|
| | pre | post | $g$ | pre | post | $g$ | pre | post | $g$ |
| ankle joint (°) | −1.3 ± 5.1 | −0.0 ± 6.2 | 0.45 | 13.7 ± 7.8 | 15.1 ± 6.0 | 0.39 | −18.0 ± 3.7 | −18.4 ± 4.4 | 0.15 |
| knee joint (°) | −3.7 ± 3.9 | −6.5 ± 6.0 | 0.53 | −11.6 ± 4.5 | −11.3 ± 4.6 | 0.05 | −32.8 ± 5.8 | −34.6 ± 5.8 | 0.41 |

experimental efficiency values provided by the paper of Hill [20], where the values are presented as a function of relative load which we then transposed to the shortening velocity (normalized to $V_{max}$) using the classical Hill equation [8]. The corresponding values of enthalpy efficiency and shortening velocity were fitted using a cubic spline, giving the right-skewed parabolic-shaped curve with a peak efficiency of 0.45 at a velocity of 0.18 $V_{max}$. The resulting function was then used to calculate the soleus efficiency during running.

## (h) Statistics

An analysis of variance for repeated measures including *post hoc* analysis (adjusted $p$-values reported) was performed for the group comparison. Anthropometric group differences as well as baseline differences of the plantar flexion moment, AT stiffness and energetic cost were tested using a *t*-test for independent samples. A paired *t*-test was used to analyse the training effects on the assessed gait characteristics, kinematics and MTU and fascicle parameters. The level of significance was set to $\alpha = 0.05$. Effect sizes (Hedges's $g$) assess the strength of the intervention effects (see electronic supplementary material for details).

## 3. Results

There were no significant differences in age ($p = 0.421$), height ($p = 0.361$) and body mass ($p = 0.382$) between the intervention and control groups. No baseline differences between groups were observed for the maximum plantar flexion moment ($p = 0.894$), AT stiffness ($p = 0.421$) and energetic cost ($p = 0.143$; table 1). Both the plantar flexion moment and AT stiffness increased significantly in the intervention

group ($p = 0.024$, $p = 0.048$) without significant changes in the controls ($p = 0.296$, $p = 0.745$; table 1). Furthermore, we found a significant decrease in the energetic cost of running following the 14 weeks of training in the intervention group ($p = 0.028$) and no significant changes in the control group ($p = 0.688$; table 1). Neither group showed any significant changes in the strike index (intervention $p = 0.868$, control $p = 0.868$), stance time ($p = 0.283$, $p = 0.283$), flight time ($p = 0.981$, $p = 0.252$) and cadence ($p = 0.310$, $p = 0.384$; table 1) after training, indicating that the training intervention did not influence the foot strike pattern.

Following the intervention, ankle and knee joint kinematics did not significantly change during the stance phase, i.e. joint angles at touchdown (ankle $p = 0.108$, knee $p = 0.064$), toe-off ($p = 0.161$, $p = 0.844$), maximal ankle dorsiflexion ($p = 0.576$) and maximal knee flexion ($p = 0.138$; table 2 and figure 2). The soleus MTU showed a lengthening–shortening behaviour during the stance phase, with shortening starting at 59 ± 2% of the stance phase similarly pre- and post-intervention ($p = 0.266$, $g = 0.30$; see the Statistics section; figure 3). The training had no effect on the MTU length, length changes and velocity, neither when averaged over the entire stance phase ($p = 0.943$, $p = 0.273$, $p = 0.274$) nor over the subphase of MTU lengthening ($p = 0.931$, $p = 0.893$, $p = 0.788$) or MTU shortening ($p = 0.946$, $p = 0.470$, $p = 0.189$; table 3 and figure 3). Despite the MTU lengthening, the soleus muscle fascicles shortened continuously throughout the entire stance phase (figure 3). Following the intervention, the fascicle shortening was not significantly different over the entire stance phase ($p = 0.662$) and the phase of MTU

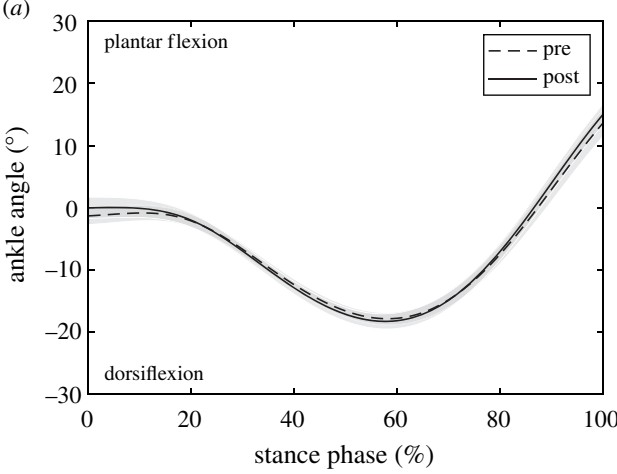

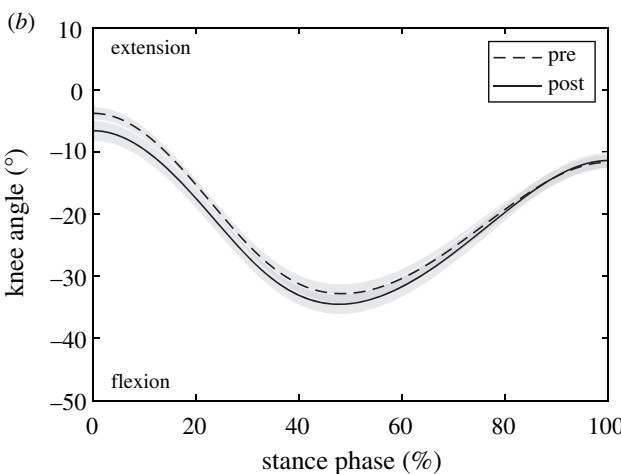

**Figure 2.** (a) Ankle joint angle and (b) knee joint angle during the stance phase of running before and after the training intervention (mean ± s.e.m., $n = 13$).

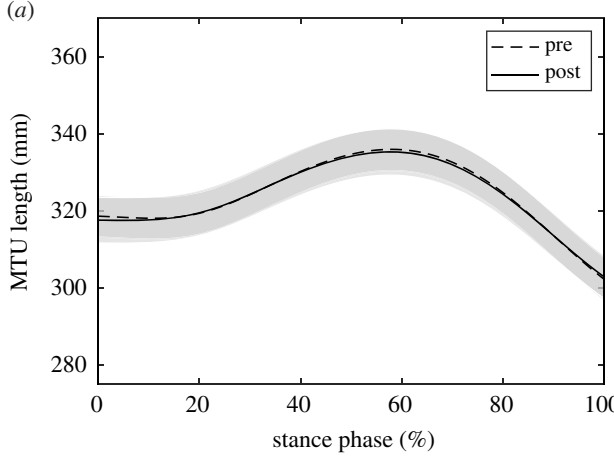

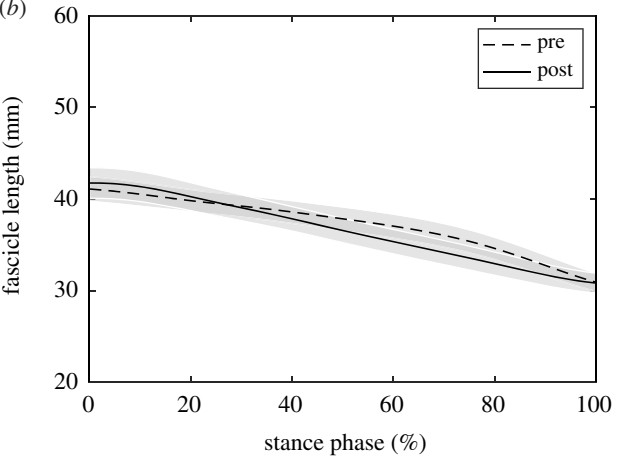

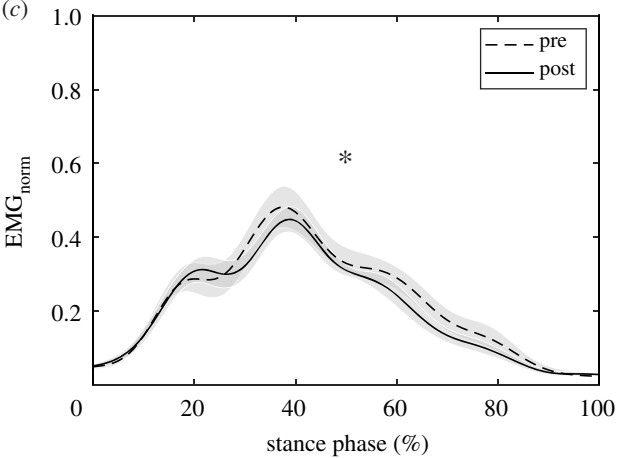

**Figure 3.** (a) Soleus MTU length, (b) muscle fascicle length and (c) EMG activity (normalized to a maximum voluntary isometric contraction, during the stance phase of running before and after the training intervention (mean ± s.e.m., $n = 13$). *Significant difference of the stance phase-averaged EMG activation between pre and post ($p < 0.05$).

lengthening ($p = 0.106$) but in the phase of MTU shortening ($p = 0.016$; table 3). $L_0$ (pre $43.1 \pm 5.7$ mm, post $44.1 \pm 8.9$ mm, $p = 0.767$, $g = 0.08$) and thus $V_{max}$ (pre $291 \pm 38$ mm s$^{-1}$, post $298 \pm 17$ mm s$^{-1}$, $p = 0.767$, $g = 0.08$) were not significantly altered due to training. The operating fascicle length averaged over the stance phase (pre $0.87 \pm 0.11 L_0$, post $0.85 \pm 0.13 L_0$, $p = 0.360$, $g = 0.16$), but also during MTU lengthening (pre $0.92 \pm 0.12 L_0$, post $0.91 \pm 0.15 L_0$, $p = 0.772$, $g = 0.07$) and shortening (pre $0.81 \pm 0.10 L_0$, post $0.76 \pm 0.11 L_0$, $p = 0.226$, $g = 0.32$), was not significantly changed following training. Consequently, the force–length potential was not significantly different between pre- and post-training in the different phases (stance $p = 0.172$, $g = 0.14$, MTU lengthening $p = 0.713$, $g = 0.10$, MTU shortening $p = 0.640$, $g = 0.12$; figure 4).

After training, the soleus force–velocity potential was significantly lower in the phase of MTU lengthening ($p = 0.030$, $g = 0.64$) and significantly higher when the MTU shortened ($p = 0.045$, $g = 0.58$) with no significant difference over the entire stance ($p = 0.249$, $g = 0.31$; figure 4). This was the consequence of a tendency towards higher fascicle shortening velocity during MTU lengthening (pre $-0.088 \pm 0.054 V_{max}$, post $-0.129 \pm 0.061 V_{max}$, $p = 0.073$, $g = 0.51$) and a significantly lower velocity during MTU shortening after training (pre $-0.174 \pm 0.057 V_{max}$, post $-0.127 \pm 0.008 V_{max}$, $p = 0.007$, $g = 0.83$). Furthermore, the averaged EMG activation over the phase of MTU shortening ($p = 0.028$, $g = 0.67$) and the entire stance phase was significantly reduced following

the intervention ($p = 0.017$, $g = 0.60$; figures 3 and 4). Compared with pre-intervention running, the fascicle velocity in the phase of MTU lengthening was closer to the velocity for optimal enthalpy efficiency after the training (figure 5). Consequently, the fascicles operated at a significantly higher enthalpy efficiency in the phase of MTU lengthening after the training ($p = 0.006$, $g = 0.85$; figures 5 and 6), while there was no significant pre–post difference in the phase of MTU shortening ($p = 0.640$, $g = 0.12$; figure 6). Over the entire stance phase of running, the efficiency of the fascicle

**Table 3.** Soleus MTU length, length changes and velocity as well as muscle fascicle length, fascicle shortening distance and fascicle velocity averaged over the phase of MTU lengthening, MTU shortening and over the entire stance phase during running before and after the training intervention (mean ± s.d., effect size $g$, $n = 13$).

| | MTU lengthening | | | MTU shortening | | | stance phase | | |
|---|---|---|---|---|---|---|---|---|---|
| | pre | post | $g$ | pre | post | $g$ | pre | post | $g$ |
| MTU length (mm) | 325 ± 20 | 325 ± 21 | 0.02 | 323 ± 20 | 323 ± 21 | 0.02 | 324 ± 20 | 324 ± 21 | 0.02 |
| MTU length changes (mm) | 18.4 ± 2.0 | 18.2 ± 3.2 | 0.04 | −33.9 ± 9.3 | −32.5 ± 5.6 | 0.19 | −16.4 ± 9.0 | −14.8 ± 5.6 | 0.30 |
| MTU velocity (mm s$^{-1}$) | 97 ± 15 | 98 ± 22 | 0.07 | −259 ± 52 | −244 ± 33 | 0.36 | −173 ± 29 | −164 ± 21 | 0.30 |
| fascicle length (mm) | 39.2 ± 4.4 | 39.0 ± 5.1 | 0.03 | 34.5 ± 4.3 | 33.1 ± 4.5 | 0.23 | 37.2 ± 4.3 | 36.5 ± 4.8 | 0.12 |
| fascicle shortening (mm) | −5.21 ± 2.68 | −6.75 ± 3.08 | 0.45 | −6.49 ± 2.02 | −4.98 ± 1.23[a] | 0.72 | −11.05 ± 3.32 | −11.53 ± 3.47 | 0.12 |
| fascicle velocity (mm s$^{-1}$) | −21.2 ± 16.7 | −33.4 ± 17.5 | 0.52 | −49.1 ± 16.7 | −35.8 ± 10.1[a] | 0.71 | −33.0 ± 10.8 | −34.6 ± 11.0 | 0.10 |

[a]Significant difference to pre ($p < 0.05$).

shortening was also significantly increased following the training ($p = 0.025$, $g = 0.66$; figure 6).

## 4. Discussion

Our current study showed for the first time that specific muscle–tendon training that increases plantar flexor muscle strength and AT stiffness facilitates the enthalpy efficiency of the soleus muscle during the stance phase of running. The increased enthalpy efficiency was found in the first part of the stance phase where the soleus muscle produces work by active shortening and transfers muscular work to the tendon as strain energy. Furthermore, the results provide additional evidence that a combination of greater plantar flexor muscle strength and AT stiffness decreases the energy cost of running [14,15] and indicates that the soleus enthalpy efficiency is a contributive determinant.

Following the intervention, the energetic cost of running was significantly reduced by about 4%, a quantity reported to be above test–retest typical errors [38] and to substantially enhance endurance running performance [39]. At the same time, the soleus, which is the main muscle for work/energy production during running [12], operated at a significantly increased (7%) enthalpy efficiency throughout the stance phase. The enthalpy efficiency quantifies the portion of energy from ATP hydrolysis used by a muscle that is converted into mechanical muscular work [21]. Enthalpy efficiency depends on the velocity of muscle shortening with a steep increase at low velocities until the peak at around 0.18 $V_{max}$ and again decreasing at higher shortening velocities [20,21]. For the whole stance phase, fascicle shortening, the force–length potential and the force–velocity potential of the soleus muscle were not significantly different before and after the intervention, indicating a similar energy production through muscular work of the soleus muscle. During the propulsion phase of running (i.e. MTU shortening), where both tendon and muscle transfer energy/work to the skeleton [19,40], the enthalpy efficiency of the operating soleus muscle was high pre- and post-intervention (94% and 93% of the maximum efficiency). By contrast, during the first part of the stance phase (i.e. MTU lengthening), where energy is transferred from the contractile element to the tendon, the efficiency was lower during pre-intervention running (77% of the maximum efficiency). The relevant part of the soleus fascicle shortening occurred during this first part of stance (59% of the entire shortening range). In combination with the high muscle activation (higher during MTU lengthening than during MTU shortening), this indicates an important energy production through muscular work during the phase of MTU lengthening.

The exercise-induced increase in plantar flexor muscle strength and AT stiffness was associated with an alteration of the operating fascicle velocity profile and a significant increase of the enthalpy efficiency of the soleus in the phase of MTU lengthening (88% of maximum), potentially improving the enthalpy efficiency of muscular work production. The significant increase of the enthalpy efficiency following training in the phase of MTU lengthening demonstrates that a substantial part of the entire muscular work was generated more economically. In the second part of the stance phase, where the MTU shortened, the high efficiency was maintained after the intervention and, further, the fascicles operated at a significantly higher force–velocity potential. This was possible due to a shift of the shortening velocity around the plateau of the efficiency–velocity curve, from the descending part before the training to the ascending part after training (figure 5), without a significant decline in the efficiency. Consequently, the overall enthalpy efficiency throughout the stance phase of each step was increased. The phase of MTU shortening was accompanied by a reduced soleus EMG activation after the intervention, and the overall EMG activity during the stance phase was significantly lower as well (12%). However, the higher maximum plantar flexion moment along with no significant changes in EMG$_{max}$ during the MVCs (pre 0.409 ± 0.114 mV, post 0.410 ± 0.092 mV, $p = 0.300$) and antagonistic co-activation (tibialis anterior EMG pre 0.034 ± 0.016 mV, post 0.034 ± 0.013 mV, $p = 0.923$) as measures for neural adaption after training strongly indicate muscle hypertrophy, resulting in a 13% increase of $F_{max}$. Therefore, the reductions in EMG activation may not correspond to a reduced active muscle volume. To examine this possibility, we calculated the average force of the soleus muscle ($F_s$) during the stance phase, adopting a 'Hill-type muscle model' as a function of the average force–length potential ($\lambda_l$), force–velocity potential ($\lambda_v$),

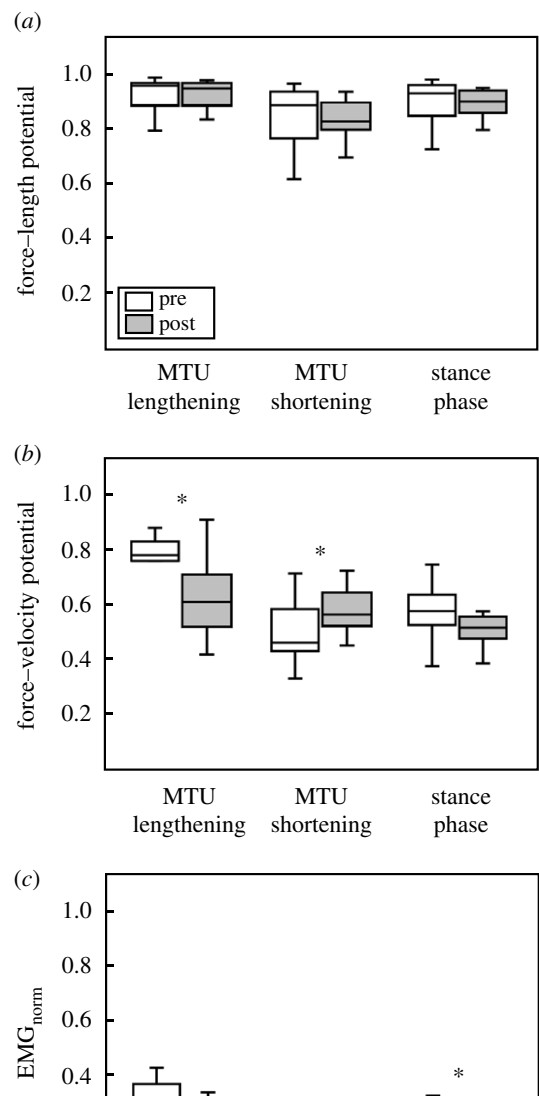

**Figure 4.** (a) Soleus fascicle force–length potential, (b) force–velocity potential and (c) EMG activity (normalized to a maximum voluntary isometric contraction, averaged over the phase of MTU lengthening, MTU shortening and the entire stance phase of running before and after the training intervention ($n = 13$). *Significant difference between pre and post ($p < 0.05$).

EMG activity ($\alpha$) and $F_{max}$ ($F_s = \lambda_l \cdot \lambda_v \cdot \alpha \cdot F_{max}$). The average force of the soleus muscle after the intervention ($F_s = 353 \pm 122$ N) did not show significant differences compared with the pre-values ($F_s = 372 \pm 112$ N, $p = 0.660$), indicating a similar active muscle volume. Similarly, the rate of muscle force generation during the stance phase ($\dot{F}_s = F_s/t_{stance}$) did not differ before ($\dot{F}_s = 1215 \pm 413$ N s$^{-1}$) and after the intervention ($\dot{F}_s = 1126 \pm 400$ N s$^{-1}$, $p = 0.498$). The above assessments suggest that the active muscle volume and the rate of muscle force generation were not the reason for the improved running economy, but rather the increase in soleus muscle operating enthalpy efficiency.

Previous studies provided evidence that the cost of force to support the body mass and the time course of force application to the ground are the major determinants of the energetic cost of running [6,41]. According to the 'cost of generating force hypothesis' [6], the rate of metabolic energy consumption is directly related to the body mass and the time available to generate force, which results in a constant cost coefficient (i.e. energy required per unit force). However, modifications in the muscle effective mechanical advantage (i.e. ratio of the muscle moment arm to the moment arm of the ground reaction force [42]) within the lower extremities can influence the cost coefficient of locomotion [43,44]. In our study, the metabolic energy cost of running was reduced after the training without any changes in the contact time and body mass, indicating a decrease of the cost coefficient. The similar strike index and lower leg kinematics before and after the intervention suggest unchanged effective mechanical advantages within the lower extremities; therefore, this would not be the reason for the reduced cost coefficient. Instead, our findings show that an adjusted time course of the soleus shortening velocity during the stance phase following the training can influence the cost coefficient as a result of increased enthalpy efficiency of the soleus and, thus, complement the earlier studies on the mechanical advantage and cost coefficient interaction [41,42]. The observed continuous soleus fascicle shortening during the stance phase is in agreement with other experiments using the ultrasound methodology and comparable running speeds [19,45]. The importance of the energy production by the plantar flexor muscles for the propulsion phase (i.e. shortening of the MTU) during running is well accepted [19,46], because the mechanical power produced at the ankle joint in this phase is highest and determines running performance [47]. Our current results regarding the enthalpy efficiency of muscular energy generation and running economy show for the first time that also the phase of the MTU lengthening is crucial for the overall metabolic energy consumption during running. Recently, findings of our group [9] but also others [48,49] provided evidence that soleus muscle dynamics may improve the economy of locomotion by a modulation of the force–length–velocity potential, thus decreasing the active muscle volume. In the present study, the soleus force–length–velocity potential throughout stance was not significantly changed following the intervention, while in the same time the adjusted time course of the shortening velocity increased the efficiency of muscle work production. Thus, the present study expands the importance of the soleus fascicle dynamics towards the efficiency–velocity dependency as a further factor for improvements of locomotor economy.

The findings of the current study provide further evidence [15,16] that strength training of the plantar flexors has the potential to enhance running economy. We used a specific high-intensity muscle–tendon training programme [24,28], targeting an adaptation of both AT stiffness and plantar flexor muscle strength [14,15], to maintain the functional integrity of the contractile and series elastic element. Strength increases without concomitant stiffening of the AT after a period of training can increase levels of operating and maximum strain [24], which have been associated with pathologies [50], and also possible functional decline [51]. On the other hand, increased stiffness without higher muscle strength may also limit function by reducing relevant operating tendon strains [51]. In our study, the maximum AT strain during the MVCs was not affected by the training (pre $6.2 \pm 1.6\%$, post $6.0 \pm 1.2\%$, $p = 0.501$) despite an increase in the plantar flexor muscle strength, indicating a balanced adaptation of muscle and tendon. Therefore, a specific

*Proc. R. Soc. B* **288**: 20202784

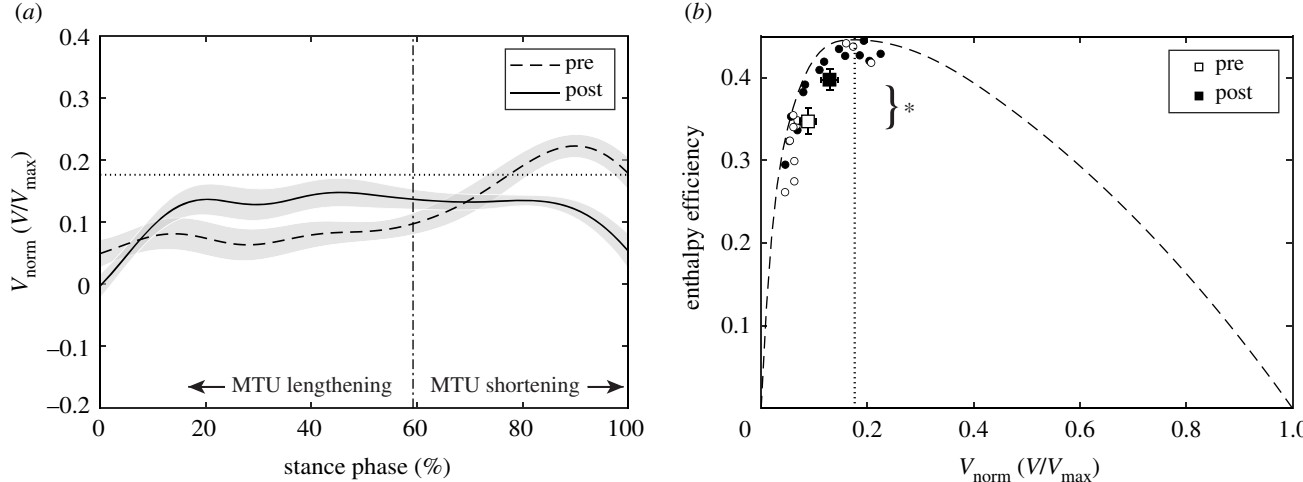

**Figure 5.** (*a*) Soleus muscle fascicle operating velocity over the stance phase of running before and after the intervention (mean ± s.e.m.) and velocity of maximum enthalpy efficiency (i.e. 0.18 $V_{max}$, horizontal dashed line). Following the intervention, the fascicle shortening velocity was closer to the velocity optimal for maximum enthalpy efficiency during most of the MTU lengthening phase. (*b*) Enthalpy efficiency–fascicle velocity relationship with average values of the phase of MTU lengthening, showing that the fascicles operated at a significantly higher enthalpy efficiency following the intervention (\**p* < 0.05). Circles indicate that the single participant values before (white) and after (black) the intervention and squares show the respective mean with standard error bars (*n* = 13). The vertical dotted line shows the velocity of maximum efficiency.

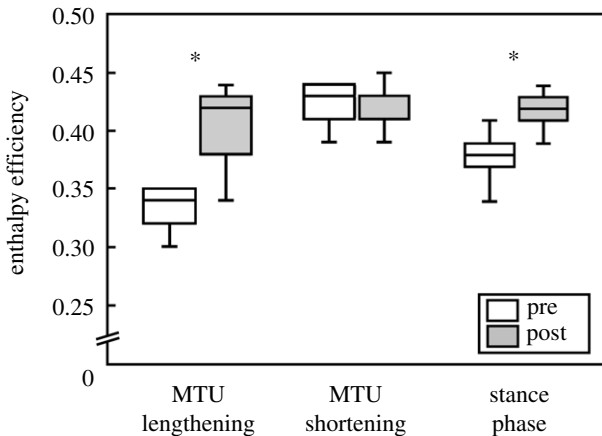

**Figure 6.** Soleus muscle fascicle enthalpy efficiency averaged over the phase of MTU lengthening, MTU shortening and the entire stance phase of running before and after the training intervention (*n* = 13). \*Significant difference between pre and post (*p* < 0.05).

muscle-tendon training [24,28] can be recommended to improve running economy.

To assess the enthalpy efficiency–shortening velocity relationship, we used a biologically founded value of $V_{max}$ (i.e. 6.77 $L_0$ s$^{-1}$). However, during submaximal running, the lower activation level and selective slow fibre-type recruitment may affect the actual relationship. Furthermore, differences in fibre-type distribution may also affect the shape of the enthalpy efficiency–shortening velocity curve [22]. We evaluated the effect of (i) decreasing $V_{max}$ by 10% intervals and (ii) replacing the underlying efficiency values measured at the frog sartorius at 0°C from Hill [20] by the data presented by Barclay *et al.* [22] for the predominantly slow fibre-type soleus mouse muscle at 21°C, comparable with the human soleus muscle. The significant pre- to post-enthalpy efficiency increase for the MTU lengthening phase and the entire stance phase persisted for values till $V_{max-30\%}$ both using the data of Hill or Barclay *et al.* (*p* < 0.05), which confirms and strengthens the observed intervention effect

(for descriptive values and *p*-values see electronic supplementary material, S2). Furthermore, since we calculated the efficiency as a function of the soleus muscle shortening velocity (adjusted for physiological temperature) and only discussed our findings in terms of percentage change, any uncertainties about the magnitude of the enthalpy efficiency would not affect our results. The soleus fascicle dynamics were not assessed in the control group because alterations were not expected with continued training habits as previously evidenced [45]. Furthermore, the controls did not show alterations in any of the assessed parameters, giving strong support for an unchanged fascicle behaviour after the intervention period.

## 5. Conclusion

In conclusion, the current study gives new insights into the soleus muscle mechanics and metabolic energetics during human running. In support of our earlier study, an exercise-induced increase of plantar flexor muscle strength and AT stiffness reduced the metabolic energy cost of running. The proposed reason for this improvement is an alteration in the soleus fascicle velocity profile throughout the stance phase, which led to a significantly higher enthalpy efficiency of the operating soleus muscle. The enthalpy efficiency was particularly increased in the phase of MTU lengthening, where the activation is high and the soleus generates an important part of the mechanical energy required for running.

**Ethics.** The ethics committee of the Humboldt-Universität zu Berlin approved the study and the participants gave written informed consent in accordance with the Declaration of Helsinki.

**Data accessibility.** The processed datasets generated and analysed during the current study are available as part of the electronic supplementary material..

**Authors' contributions.** S.B., F.M., A.S. and A.A. designed research. S.B., F.M. and A.S. performed research. S.B. analysed data. S.B. and A.A. drafted the manuscript. F.M. and A.S. made important intellectual contributions during revision.

**Competing interests.** We declare we have no competing interests.

**Funding.** Funding for this research was supplied by the German Federal Institute of Sport Science (grant no. ZMVI14-070604/ 17-18).

**Acknowledgements.** We acknowledge the support of Antonis Ekizos, Arno Schroll, Leon Brüll and Victor Munoz-Martel for data recording and analysis.

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
