## [Reviewer comments · Proceedings of the Royal Society B: Biological Sciences]

Review History

RSPB-2020-2784.R0 (Original submission)

Review form: Reviewer 1 (Richard Blagrove)

Recommendation

Accept with minor revision (please list in comments)

Scientific importance: Is the manuscript an original and important contribution to its field?

Excellent

General interest: Is the paper of sufficient general interest?

Good

Quality of the paper: Is the overall quality of the paper suitable?

Excellent

Is the length of the paper justified?

Yes

Should the paper be seen by a specialist statistical reviewer?

No

Do you have any concerns about statistical analyses in this paper? If so, please specify them explicitly in your report.

No

It is a condition of publication that authors make their supporting data, code and materials available - either as supplementary material or hosted in an external repository. Please rate, if applicable, the supporting data on the following criteria.

Is it accessible?

N/A

Is it clear?

N/A

Is it adequate?

N/A

Do you have any ethical concerns with this paper?

No

Comments to the Author

General comments:

Many thanks for the invitation to review this paper, it was a thoroughly enjoyable and fascinating read. The paper describes the results of a 14-week muscle-tendon strength training intervention that found enhancements in running economy, plantar flexion strength, and Achilles stiffness compared to a control group. An improvement in enthalpy efficiency of the soleus muscle reveals novel insight into the mechanism by which strength training may have a positive effect of the metabolic cost of running.

In my opinion, this study is much needed in this area of research. Papers have speculated in the past around the mechanisms of change associated with improved running economy following a strength training intervention (e.g. Fletcher and Macintosh, 2017, doi: 10.3389/fphys.2017.00433; Blagrove et al., 2018, doi: 10.1007/s40279-017-0835-7), however measuring changes to the intrinsic behaviour of muscles is difficult. This study makes a very good attempt at providing that insight for the soleus. I have some minor comments that I hope will improve clarity and readability of the paper, but overall, I feel that this paper will be of considerable interest to both scientists and applied practitioners.

Specific comments:

Keywords: These should be different to the terms in the study title to enable wider search returns. Please amend. Can I suggest 'calf' 'triceps surae' 'endurance running' 'strength training'?

Line 44: 'for' should read 'in'

Line 105: Was allocation to groups completely random or were participants matched for running economy and randomised by matched pairs (or similar) to ensure minimal differences existed between groups at baseline?

Line 106: Was the participants only sport/exercise running? It would be useful for others (particularly those undertaking reviews and meta-analyses in this area) to be able to accurately determine if participants were trained 'runners' or simply people that ran as a small part of a wider exercise/sport training routine.

Line 106: Please define 'severe' in brackets here (i.e. days/weeks away from running with injury)

Line 108: Why were only rear-foot striking runners considered?

In female participants, was the menstrual cycle accounted for or hormonal contraceptive use during recruitment and testing?

A criticism often levelled at studies in the area of strength training for endurance athletes is that studies rarely equate the total amount of physical exercise done between groups, i.e. the control group do not have 'placebo' exercise(s) or add running training to match the duration of strength work performed by the intervention group (e.g. Dankel et al., 2017, doi: 10.1080/02640414.2017.1398884). Although a performance measure was not taken in this study, how do the authors know that the change in running economy they observed is not due to differences in the amount of physical training performed? An alternative, in practice, for runners could be to add running training instead of strength training to their routine, which may produce even larger improvements in economy.

The changes in soleus fascicle behaviour were not quantified in the control group. I am slightly puzzled why not. Would the authors consider this a limitation of the study?

Exercise protocol: Given that a single strength training exercise was used in the intervention I would strongly recommend that authors include an image of the exercise apparatus and set-up. I appreciate there are currently a high number of figures included but I would contend this is important for both scientific replication and applied practice.

Line 147: Why was 2.5 m/s used as the speed for all participants? Was this sufficiently slow enough to ensure a plateau in oxygen consumption and RER value of <1 during the collection period?

Line 149: The citation here is a paper comparing methods of quantifying energy cost of running. It is not clear which method was used without referring to the supplementary material.

Line 205: Which post-hoc adjustment was used?

Line 210: How were the effect sizes interpreted?

Line 229: There appears to be a word missing in this sentence. 'an altered lengthening-shortening behaviour' or similar

Line 272: It would be more accurate to discuss the change in economy in the context of within-participant variability (measurement error), rather than between-participant variability, which depends on the sample. A subtle tweak to wording and the reference (eg Blagrove et al., 2017, doi: 10.1080/17461391.2017.1364301; Shaw et al., 2013, doi: 10.1139/apnm-2013-0055) here would provide a more compelling that the 4% improvement is indeed real.

Line 304-305: Why does the higher maximum plantar flexion moment indicate hypertrophy has occurred? It would be unusual to expect substantial hypertrophy with short-duration isometric contractions. Why can the improvements in strength not be explained as neural adaptation? If so, the discussion below this statement will need to be amended.

Line 346: 'a' seems to be a typographical error here.

Line 346: The ref. 16 study (Fletcher et al., 2010) did not find a significant change in running economy following a calf strengthening intervention.

Line 350: There appears to be a word missing between 'training' and 'may'

Line 355: 'endurance performance' should read 'running economy' here as no performance measures were taken.

It has long been recognised that the soleus possesses a high proportion of slow twitch muscle fibres compared to other muscle groups (eg Gollnick et al., 1974, doi: 10.1007/BF00587415). Clearly it is possible to make the soleus stronger and given its role in locomotion and energy cost during exercise, it would certainly make sense for runners to strengthen the muscle. However, do authors think that the soleus has a limited capacity to improve its maximal force output due to its morphological characteristics? The intervention applied here would certainly be novel for the participants, thus beneficial, but would long-term engagement with this type of training for soleus continue to yield benefits in running economy?

Review form: Reviewer 2

Recommendation

Major revision is needed (please make suggestions in comments)

Scientific importance: Is the manuscript an original and important contribution to its field?

Good

General interest: Is the paper of sufficient general interest?

Good

Quality of the paper: Is the overall quality of the paper suitable?

Good

Is the length of the paper justified?

Yes

Should the paper be seen by a specialist statistical reviewer?

No

Do you have any concerns about statistical analyses in this paper? If so, please specify them explicitly in your report.

No

It is a condition of publication that authors make their supporting data, code and materials available - either as supplementary material or hosted in an external repository. Please rate, if applicable, the supporting data on the following criteria.

Is it accessible?

Yes

Is it clear?

Yes

Is it adequate?

Yes

Do you have any ethical concerns with this paper?

No

Comments to the Author

In this study, the authors examined the effects of a resistance training program on running economy, and additionally examined how changes in running economy were associated with changes in estimated soleus muscle strength, Achilles tendon stiffness, and operating soleus muscle efficiency, force-length, and force-velocity behaviour. This study provides insight into the mechanisms that may underly improvements in running economy with resistance training. The majority of our understanding of the role of series elasticity on efficiency is from controlled in situ or simulation studies. Thus, this study also provides novel insight into the implications of in vivo muscle and tendon properties during real-world conditions.

This manuscript is well-written and interesting to read, and the methods appear sound and appropriate for addressing the research questions. I only have a few comments below that aim to clarify details of the methodology and interpretation of the results.

Comments:

1. Lines 137-144: Given that increased plantar flexor strength and tendon stiffness are identified as a possible mechanism underlying the main results of this paper, it would be helpful to provide further details of how these variables were measured rather than referring readers to other papers.

For example, in Supplementary material 1, section 2:

“Furthermore, the contribution of the antagonistic muscles to the ankle joint moment was considered by means of an EMG-based method [4].” What specific method was this?

“which was determined using the tendon-excursion method [5,6] and corrected for tendon alignment during the contraction [7].” How were the moment arms corrected for tendon alignment?

“The MTJ displacement artefacts due to an unavoidable change in the ankle joint angle during the MVCs was corrected [8] and the five contractions were averaged to give a reliable measure of the elongation [9]. The AT stiffness was calculated between 50% and 100% of the maximum tendon force using linear regression [10]” How were the changes in ankle joint angle corrected?

Currently the reader would have to consult a range of other papers to fully understand the methods and their justification. More details of these methods and less reliance on previous works would be beneficial.

2. Similar to 1., given that running economy is an important variable in this paper, further details in the main text would be helpful. Since the section “Energetic cost of running” in supplementary material 1 is only one paragraph long, could this not be included in the methods section of the main text? I realize the authors may be limited in terms of length; however, these details are important for interpreting the results of this paper. Similarly, at least the first paragraph of the section “Statistics” in supplementary material 1 could be included in the main text. Important methods that could affect interpretation of results and conclusions should be easy for readers to access in the main text.

3. Line 194: Why did the authors use an efficiency-velocity function rather than a more established metabolic power function (e.g. Minetti & Alexander, 1997 or Umberger, 2010, etc.)? Mechanical work and metabolic cost depend on factors other than just velocity, so why is an efficiency function that depends only on velocity, instead of separately estimating mechanical work and metabolic cost that depend on muscle velocity, length, activation, etc., appropriate for this study? Further explanation/justification in the text would be helpful. Also, the fitted values in Table 1 of Hill (1967) are for frog muscle at 0 degrees C. Since frogs are ectotherms, the muscle temperature would be near that of the external environment, far below physiological temperature for human muscle. This could affect both muscle force and velocity (see James, 2013 for review) and therefore the fitted function. Additionally, amphibian muscle contains larger concentrations of parvalbumin compared to terrestrial muscles, which can alter the heat rate and estimated metabolic cost (Woledge et al., 1985, pp. 257-260). What are the implications of these

considerations on the results of this study?

James, R. S. (2013). A review of the thermal sensitivity of the mechanics of vertebrate skeletal muscle. *Journal of Comparative Physiology B*, 183(6), 723-733.

Woledge, R. C., Curtin, N. A., & Homsher, E. (1985). Energetic aspects of muscle contraction. *Monographs of the Physiological Society*.

4. Line 268: "... the results provide additional evidence that a combination of greater plantar flexor muscle strength and Achilles tendon stiffness decrease the energy cost of running [14,15] and indicate that the soleus enthalpy efficiency is a contributive determinant." It's alluded to with "a combination" but consider an additional sentence here noting that an increase in stiffness by itself may not increase efficiency. Later in line 349 the authors state "strength increases without concomitant stiffening of the AT after a period of training may increase levels of operating and maximum AT strain [24], which have been associated with pathologies [53] but also possible functional decline [54]." Function may also decline with increases in stiffness without concomitant increases in muscle strength. For example, see Figure 5 in Lichtwark and Wilson (2005) in which muscle efficiency during running decreased with increases in AT stiffness beyond the optimal stiffness.

Lichtwark, G. A., & Wilson, A. M. (2007). Is Achilles tendon compliance optimised for maximum muscle efficiency during locomotion? *Journal of Biomechanics*, 40(8), 1768-1775.

5. Line 291: "The exercise-induced increase in muscle strength and AT stiffness resulted in an alteration of the operating fascicle velocity profile that led to a significant increase of the enthalpy efficiency of the operating soleus [...], improving the efficiency of muscular work production." Because the only factor that was manipulated in this study was the exercise intervention, changes in muscle strength, AT stiffness, fascicle velocities, and enthalpy efficiency are only associated with one another rather than there being any causal relationship between them.

Decision letter (RSPB-2020-2784.R0)

07-Dec-2020

Dear Dr Bohm:

Your manuscript has now been peer reviewed and the reviews have been assessed by an Associate Editor. The reviewers' comments (not including confidential comments to the Editor) and the comments from the Associate Editor are included at the end of this email for your reference. As you will see, the reviewers and the Editors have raised some concerns with your manuscript and we would like to invite you to revise your manuscript to address them.

Research ethics:

Use of animals and field studies:

It is a condition of publication that you make available the data and research materials supporting the results in the article. Please see our Data Sharing Policies (<https://royalsociety.org/journals/authors/author-guidelines/#data>). Datasets should be deposited in an appropriate publicly available repository and details of the associated accession number, link or DOI to the datasets must be included in the Data Accessibility section of the article (<https://royalsociety.org/journals/ethics-policies/data-sharing-mining/>). Reference(s) to datasets should also be included in the reference list of the article with DOIs (where available).

Please note: your data accessibility states "Analyses of experimental data were performed using MATLAB (R2016b, MathWorks) and the code is available from the corresponding author on reasonable request."-- this is not acceptable via our policy. All data and code must be provided with your revised MS or we may delay or reject your study unless this is amended. For more information please see our open data policy <http://royalsocietypublishing.org/data-sharing>.

All supplementary materials accompanying an accepted article will be treated as in their final form. They will be published alongside the paper on the journal website and posted on the online figshare repository. Files on figshare will be made available approximately one week before the

accompanying article so that the supplementary material can be attributed a unique DOI. Please try to submit all supplementary material as a single file.

Please submit a copy of your revised paper within three weeks. If we do not hear from you within this time your manuscript will be rejected. If you are unable to meet this deadline please let us know as soon as possible, as we may be able to grant a short extension.

Best wishes,
Dr John Hutchinson, Editor
mailto:proceedingsb@royalsociety.org

Associate Editor
Board Member: 1
Comments to Author:

Dear Dr. Bohm,

Thank you for submitting your manuscript entitled "Enthalpy efficiency of the soleus muscle contributes to improvements in running economy" to the Proceedings of the Royal Society. I have received two peer reviews, and both are highly supportive of your manuscript but also have a several suggestions, which I hope you will find useful when revising your manuscript.

Proceedings B aims to publish studies that significantly increase or alter our current understandings in a way that is relevant to a broad readership beyond the disciplinary area of the manuscript. Both reviewers find your study of high scientific importance and broad interest, and many of their comments aim mainly at improving the clarity of the manuscript. The reviewers furthermore ask for additional information on the methodology and share their thoughts concerning the findings, cautioning against overstatements and arguing for nuance.

Reviewer(s)' Comments to Author:

Referee: 1

Comments to the Author(s)
General comments:

Many thanks for the invitation to review this paper, it was a thoroughly enjoyable and fascinating read. The paper describes the results of a 14-week muscle-tendon strength training intervention that found enhancements in running economy, plantar flexion strength, and Achilles stiffness compared to a control group. An improvement in enthalpy efficiency of the soleus muscle reveals novel insight into the mechanism by which strength training may have a positive effect of the metabolic cost of running.

In my opinion, this study is much needed in this area of research. Papers have speculated in the past around the mechanisms of change associated with improved running economy following a strength training intervention (e.g. Fletcher and Macintosh, 2017, doi: 10.3389/fphys.2017.00433; Blagrove et al., 2018, doi: 10.1007/s40279-017-0835-7), however measuring changes to the intrinsic behaviour of muscles is difficult. This study makes a very good attempt at providing that insight for the soleus. I have some minor comments that I hope will improve clarity and readability of

the paper, but overall, I feel that this paper will be of considerable interest to both scientists and applied practitioners.

Specific comments:

Keywords: These should be different to the terms in the study title to enable wider search returns. Please amend. Can I suggest 'calf' 'triceps surae' 'endurance running' 'strength training'?

Line 44: 'for' should read 'in'

Line 105: Was allocation to groups completely random or were participants matched for running economy and randomised by matched pairs (or similar) to ensure minimal differences existed between groups at baseline?

Line 106: Was the participants only sport/exercise running? It would be useful for others (particularly those undertaking reviews and meta-analyses in this area) to be able to accurately determine if participants were trained 'runners' or simply people that ran as a small part of a wider exercise/sport training routine.

Line 106: Please define 'severe' in brackets here (i.e. days/weeks away from running with injury)

Line 108: Why were only rear-foot striking runners considered?

In female participants, was the menstrual cycle accounted for or hormonal contraceptive use during recruitment and testing?

A criticism often levelled at studies in the area of strength training for endurance athletes is that studies rarely equate the total amount of physical exercise done between groups, i.e. the control group do not have 'placebo' exercise(s) or add running training to match the duration of strength work performed by the intervention group (e.g. Dankel et al., 2017, doi: 10.1080/02640414.2017.1398884). Although a performance measure was not taken in this study, how do the authors know that the change in running economy they observed is not due to differences in the amount of physical training performed? An alternative, in practice, for runners could be to add running training instead of strength training to their routine, which may produce even larger improvements in economy.

The changes in soleus fascicle behaviour were not quantified in the control group. I am slightly puzzled why not. Would the authors consider this a limitation of the study?

Exercise protocol: Given that a single strength training exercise was used in the intervention I would strongly recommend that authors include an image of the exercise apparatus and set-up. I appreciate there are currently a high number of figures included but I would contend this is important for both scientific replication and applied practice.

Line 147: Why was 2.5 m/s used as the speed for all participants? Was this sufficiently slow enough to ensure a plateau in oxygen consumption and RER value of <1 during the collection period?

Line 149: The citation here is a paper comparing methods of quantifying energy cost of running. It is not clear which method was used without referring to the supplementary material.

Line 205: Which post-hoc adjustment was used?

Line 210: How were the effect sizes interpreted?

Line 229: There appears to be a word missing in this sentence. 'an altered lengthening-shortening behaviour' or similar

Line 272: It would be more accurate to discuss the change in economy in the context of within-participant variability (measurement error), rather than between-participant variability, which depends on the sample. A subtle tweak to wording and the reference (eg Blagrove et al., 2017, doi; 10.1080/17461391.2017.1364301; Shaw et al., 2013, doi: 10.1139/apnm-2013-0055) here would provide a more compelling that the 4% improvement is indeed real.

Line 304-305: Why does the higher maximum plantar flexion moment indicate hypertrophy has occurred? It would be unusual to expect substantial hypertrophy with short-duration isometric contractions. Why can the improvements in strength not be explained as neural adaptation? If so, the discussion below this statement will need to be amended.

Line 346: 'a' seems to be a typographical error here.

Line 346: The ref. 16 study (Fletcher et al., 2010) did not find a significant change in running economy following a calf strengthening intervention.

Line 350: There appears to be a word missing between 'training' and 'may'

Line 355: 'endurance performance' should read 'running economy' here as no performance measures were taken.

It has long been recognised that the soleus possesses a high proportion of slow twitch muscle fibres compared to other muscle groups (eg Gollnick et al., 1974, doi: 10.1007/BF00587415). Clearly it is possible to make the soleus stronger and given its role in locomotion and energy cost during exercise, it would certainly make sense for runners to strengthen the muscle. However, do authors think that the soleus has a limited capacity to improve its maximal force output due to its morphological characteristics? The intervention applied here would certainly be novel for the participants, thus beneficial, but would long-term engagement with this type of training for soleus continue to yield benefits in running economy?

Referee: 2

Comments to the Author(s)

In this study, the authors examined the effects of a resistance training program on running economy, and additionally examined how changes in running economy were associated with changes in estimated soleus muscle strength, Achilles tendon stiffness, and operating soleus muscle efficiency, force-length, and force-velocity behaviour. This study provides insight into the mechanisms that may underly improvements in running economy with resistance training. The majority of our understanding of the role of series elasticity on efficiency is from controlled in situ or simulation studies. Thus, this study also provides novel insight into the implications of in vivo muscle and tendon properties during real-world conditions.

This manuscript is well-written and interesting to read, and the methods appear sound and appropriate for addressing the research questions. I only have a few comments below that aim to clarify details of the methodology and interpretation of the results.

Comments:

1. Lines 137-144: Given that increased plantar flexor strength and tendon stiffness are identified as a possible mechanism underlying the main results of this paper, it would be helpful to provide further details of how these variables were measured rather than referring readers to other papers.

For example, in Supplementary material 1, section 2:

“Furthermore, the contribution of the antagonistic muscles to the ankle joint moment was considered by means of an EMG-based method [4].” What specific method was this?

“which was determined using the tendon-excursion method [5,6] and corrected for tendon alignment during the contraction [7].” How were the moment arms corrected for tendon alignment?

“The MTJ displacement artefacts due to an unavoidable change in the ankle joint angle during the MVCs was corrected [8] and the five contractions were averaged to give a reliable measure of the elongation [9]. The AT stiffness was calculated between 50% and 100% of the maximum tendon force using linear regression [10]” How were the changes in ankle joint angle corrected?

Currently the reader would have to consult a range of other papers to fully understand the methods and their justification. More details of these methods and less reliance on previous works would be beneficial.

2. Similar to 1., given that running economy is an important variable in this paper, further details in the main text would be helpful. Since the section “Energetic cost of running” in supplementary material 1 is only one paragraph long, could this not be included in the methods section of the main text? I realize the authors may be limited in terms of length; however, these details are important for interpreting the results of this paper. Similarly, at least the first paragraph of the section “Statistics” in supplementary material 1 could be included in the main text. Important methods that could affect interpretation of results and conclusions should be easy for readers to access in the main text.

3. Line 194: Why did the authors use an efficiency-velocity function rather than a more established metabolic power function (e.g. Minetti & Alexander, 1997 or Umberger, 2010, etc.)? Mechanical work and metabolic cost depend on factors other than just velocity, so why is an efficiency function that depends only on velocity, instead of separately estimating mechanical work and metabolic cost that depend on muscle velocity, length, activation, etc., appropriate for this study? Further explanation/justification in the text would be helpful. Also, the fitted values in Table 1 of Hill (1967) are for frog muscle at 0 degrees C. Since frogs are ectotherms, the muscle temperature would be near that of the external environment, far below physiological temperature for human muscle. This could affect both muscle force and velocity (see James, 2013 for review) and therefore the fitted function. Additionally, amphibian muscle contains larger concentrations of parvalbumin compared to terrestrial muscles, which can alter the heat rate and estimated metabolic cost (Woledge et al., 1985, pp. 257-260). What are the implications of these considerations on the results of this study?

James, R. S. (2013). A review of the thermal sensitivity of the mechanics of vertebrate skeletal muscle. *Journal of Comparative Physiology B*, 183(6), 723-733.

Woledge, R. C., Curtin, N. A., & Homsher, E. (1985). Energetic aspects of muscle contraction. *Monographs of the Physiological Society*.

4. Line 268: “... the results provide additional evidence that a combination of greater plantar flexor muscle strength and Achilles tendon stiffness decrease the energy cost of running [14,15] and indicate that the soleus enthalpy efficiency is a contributive determinant.” It’s alluded to with “a combination” but consider an additional sentence here noting that an increase in stiffness by itself may not increase efficiency. Later in line 349 the authors state “strength increases without concomitant stiffening of the AT after a period of training may increase levels of operating and maximum AT strain [24], which have been associated with pathologies [53] but also possible functional decline [54].” Function may also decline with increases in stiffness without concomitant increases in muscle strength. For example, see Figure 5 in Lichtwark and Wilson (2005) in which muscle efficiency during running decreased with increases in AT stiffness beyond the optimal stiffness.

Lichtwark, G. A., & Wilson, A. M. (2007). Is Achilles tendon compliance optimised for maximum muscle efficiency during locomotion? *Journal of Biomechanics*, 40(8), 1768-1775.

5. Line 291: "The exercise-induced increase in muscle strength and AT stiffness resulted in an alteration of the operating fascicle velocity profile that led to a significant increase of the enthalpy efficiency of the operating soleus [...], improving the efficiency of muscular work production." Because the only factor that was manipulated in this study was the exercise intervention, changes in muscle strength, AT stiffness, fascicle velocities, and enthalpy efficiency are only associated with one another rather than there being any causal relationship between them.

Author's Response to Decision Letter for (RSPB-2020-2784.R0)

See Appendix A.

RSPB-2020-2784.R1 (Revision)

Review form: Reviewer 1 (Richard Blagrove)

Recommendation

Accept as is

Scientific importance: Is the manuscript an original and important contribution to its field?

Excellent

General interest: Is the paper of sufficient general interest?

Excellent

Quality of the paper: Is the overall quality of the paper suitable?

Excellent

Is the length of the paper justified?

Yes

Should the paper be seen by a specialist statistical reviewer?

No

Do you have any concerns about statistical analyses in this paper? If so, please specify them explicitly in your report.

No

It is a condition of publication that authors make their supporting data, code and materials available - either as supplementary material or hosted in an external repository. Please rate, if applicable, the supporting data on the following criteria.

Is it accessible?

Yes

Is it clear?

Yes

Is it adequate?

Yes

Do you have any ethical concerns with this paper?

No

Comments to the Author

Many thanks for taking the time to provide clear and comprehensive responses to my comments and questions. I am satisfied they have been appropriately addressed. I look forward to seeing this paper published and will circulate it to my networks. I'm sure i'll refer to it regularly.

Decision letter (RSPB-2020-2784.R1)

05-Jan-2021

Dear Dr Bohm

I am pleased to inform you that your manuscript entitled "Enthalpy efficiency of the soleus muscle contributes to improvements in running economy" has been accepted for publication in Proceedings B. Congratulations!!

Open Access

You are invited to opt for Open Access, making your freely available to all as soon as it is ready for publication under a CCBY licence. Our article processing charge for Open Access is £1700. Corresponding authors from member institutions (<http://royalsocietypublishing.org/site/librarians/allmembers.xhtml>) receive a 25% discount to these charges. For more information please visit <http://royalsocietypublishing.org/open-access>.

Paper charges

Sincerely,
Dr John Hutchinson
Editor, Proceedings B
mailto: proceedingsb@royalsociety.org

Associate Editor:
Board Member: 1
Comments to Author:
(There are no comments.)

Appendix A

Referee: 1

General comments:

Comment: Many thanks for the invitation to review this paper, it was a thoroughly enjoyable and fascinating read. The paper describes the results of a 14-week muscle-tendon strength training intervention that found enhancements in running economy, plantar flexion strength, and Achilles stiffness compared to a control group. An improvement in enthalpy efficiency of the soleus muscle reveals novel insight into the mechanism by which strength training may have a positive effect of the metabolic cost of running.

In my opinion, this study is much needed in this area of research. Papers have speculated in the past around the mechanisms of change associated with improved running economy following a strength training intervention (e.g. Fletcher and Macintosh, 2017, doi: 10.3389/fphys.2017.00433; Blagrove et al., 2018, doi: 10.1007/s40279-017-0835-7), however measuring changes to the intrinsic behaviour of muscles is difficult. This study makes a very good attempt at providing that insight for the soleus. I have some minor comments that I hope will improve clarity and readability of the paper, but overall, I feel that this paper will be of considerable interest to both scientists and applied practitioners.

Response: Thank you for your thorough and valuable comments.

Specific comments:

Comment: Keywords: These should be different to the terms in the study title to enable wider search returns. Please amend. Can I suggest 'calf' 'triceps surae' 'endurance running' 'strength training'?

Response: Thanks for this comment. We replaced four of the keywords to: "*Force-length and force-velocity relationship, enthalpy-velocity relationship, triceps surae, endurance running, strength training, tendon stiffness*"

Comment: Line 44: 'for' should read 'in'

Response: Corrected. Thank you.

Comment: Line 105: Was allocation to groups completely random or were participants matched for running economy and randomised by matched pairs (or similar) to ensure minimal differences existed between groups at baseline?

Response: It was completely random and the outcome parameters for the group comparison were not significantly different at the baseline level as reported.

Comment: Line 106: Was the participants only sport/exercise running? It would be useful for others (particularly those undertaking reviews and meta-analyses in this area) to be able to accurately determine if participants were trained 'runners' or simply people that ran as a small part of a wider exercise/sport training routine.

Response: Participants were running on a regular, yet recreational basis with a minimum of twice a week set as an inclusion criterion. None of the participants was involved in professional competitive running. We added to the following information to be more clear (page: 3, line: 103):

“Inclusion criteria were age 20 to 40 years, at least two running sessions weekly on a recreational basis and no muscular-tendinous injuries in the previous year.”

Comment: Line 106: Please define ‘severe’ in brackets here (i.e. days/weeks away from running with injury)

Response: ‘Severe’ in the context meant potential injuries that affected the running habits of the participants in the past year. We deleted the word to avoid confusion.

Comment: Line 108: Why were only rear-foot striking runners considered?

Response: There is an ongoing debate on the effect on foot strike pattern on running economy in the scientific community. Thus, to avoid any potential confounding effects on our study outcomes we excluded this factor by recruiting a homogenous group of rear foot runners, which is also by far the most common strike pattern (1,2,3,4). We added to the following information to be more clear (page: 3, line: 108):

“Only habitual rearfoot-striking runners were considered because it is the most common foot strike pattern (4) and also to avoid potential confounding effects of the strike pattern on our study outcomes. To quantify the foot strike pattern [...]”

References:

1. Kasmer ME, Liu X-C, Roberts KG, Valadao JM. 2013 Foot-strike pattern and performance in a marathon. *Int J Sports Physiol Perform* **8**, 286–292.
2. Patoz A, Lussiana T, Gindre C, Hébert-Losier K. 2019 Recognition of Foot Strike Pattern in Asian Recreational Runners. *Sports (Basel)* **7**.
3. Cheung RTH, Wong RYL, Chung TKW, Choi RT, Leung WWY, Shek DHY. 2017 Relationship between foot strike pattern, running speed, and footwear condition in recreational distance runners. *Sports Biomech* **16**, 238–247.
4. Santuz A, Ekizos A, Arampatzis A. 2016 A Pressure Plate-Based Method for the Automatic Assessment of Foot Strike Patterns During Running. *Ann Biomed Eng* **44**, 1646–1655.

Comment: In female participants, was the menstrual cycle accounted for or hormonal contraceptive use during recruitment and testing?

Response: Menstrual cycle status was not considered systematically because of the difficulty to align pre/post measurements sessions and menstrual cycle time points. However, three of the four females of the 13 participants of the intervention group self-reported their cycle status; female 1: pre: early follicular phase (day 3 of cycle) and post: late luteal phase (day 26), female 2: pre: late follicular phase (day 9) and post: early luteal phase (day 16) and female 3: pre: late luteal (day 26) and post: late follicular phase (day 9). Over the time course of the menstrual cycle only the mid-luteal phase has been reported to impair running economy (2,3). Note that none of the three females reported this particular phase of the menstrual cycle during the test sessions. With regard to contraceptives, from the four females only two used a hormone spiral as contraceptive. Such low-dose contraceptives have been suggested to not interfere significantly with running economy (1). However, the specific application of hormone spirals in the context of running economy seems not investigated well to the best of our knowledge. Furthermore, when performing a sensitivity analysis of our reported intervention effect by changing the post energy cost values of the four females, we found that the intervention effect on the

energetic cost would remain significant ($p < 0.05$) in case of up to a ~4% higher post value in all of the four females at the same time.

Therefore, considering a) the low number of females in the intervention group, b) that none of the females reported critical mid-luteal phase during testing, c) only two used low-dose contraceptives and d) a quite robust intervention effect against the unlikely case that all the four females would have a higher energetic cost during the post test, we can argue that our found improvement in running economy following training is likely not affected by these factors.

References:

1. Rebelo ACS, Zuttin RS, Verlengia R, Cesar M de C, de Sá MFS, da Silva E. 2010 Effect of low-dose combined oral contraceptive on aerobic capacity and anaerobic threshold level in active and sedentary young women. *Contraception* **81**, 309–315.
2. Goldsmith E, Glaister M. 2020 The effect of the menstrual cycle on running economy. *J Sports Med Phys Fitness* **60**, 610–617.
3. Williams TJ, Krahenbuhl GS. 1997 Menstrual cycle phase and running economy. *Med Sci Sports Exerc* **29**, 1609–1618.

Comment: A criticism often levelled at studies in the area of strength training for endurance athletes is that studies rarely equate the total amount of physical exercise done between groups, i.e. the control group do not have 'placebo' exercise(s) or add running training to match the duration of strength work performed by the intervention group (e.g. Dankel et al., 2017, doi: 10.1080/02640414.2017.1398884). Although a performance measure was not taken in this study, how do the authors know that the change in running economy they observed is not due to differences in the amount of physical training performed? An alternative, in practice, for runners could be to add running training instead of strength training to their routine, which may produce even larger improvements in economy.

Response: Thank you for this important comment. We did not include an additional group that performed a time-matched running training in our experimental design. The reason was that in an earlier study (1) we applied specifically running training (i.e. focusing on the alteration of the running technique) in a group of experienced runners and we did not find any effects on running economy after the 14 weeks of running training (3x/week 30min). Furthermore, several studies in the past have shown that running training itself does not improve running economy (ref. 2: 6 weeks, ~160 km,) particularly in trained runners as in our study (ref. 3: 8 weeks, 210 km added). Therefore, we are confident that the specific strength training, which improved muscle strength of the plantar flexors and Achilles tendon stiffness, was the reason for the improved running economy and that running training in experienced runners cannot cause any additional improvements in running economy.

References:

1. Ekizos A, Santuz A, Arampatzis A. 2018 Short- and long-term effects of altered point of ground reaction force application on human running energetics. *Journal of Experimental Biology* **221**, jeb176719.
2. Daniels JT, Yarbrough RA, Foster C. 1978 Changes in VO₂ max and running performance with training. *Eur J Appl Physiol Occup Physiol* **39**, 249–254.
3. Lake MJ, Cavanagh PR. 1996 Six weeks of training does not change running mechanics or improve running economy. *Medicine & Science in Sports & Exercise* **28**, 860–869.

Comment: The changes in soleus fascicle behavior were not quantified in the control group. I am slightly puzzled why not. Would the authors consider this a limitation of the study?

Response: Thank you for the comment. The participants of the control group did not change their regular training habits and therefore changes in the fascicle dynamics were not expected. In this regard, Werkhausen et al. (2019) did not find any alterations in the fascicle behavior of the soleus and gastrocnemius medialis after 10 weeks. Furthermore, our control group did not show any changes in maximal ankle joint moment and tendon stiffness, running kinematics, temporal gait characteristics, foot strike pattern and energetic cost after the intervention period. Together, this gives strong support for an unchanged fascicle behavior of soleus after our intervention period in the controls and consequently we

would not see this as a limitation. According to the reviewer's suggestion, we noted this issue in the limitation part of the revised manuscript as follows (page: 9, line: 360):

“The soleus fascicle dynamics were not assessed in the control group because alterations were not expected with continued training habits as previously evidenced (1). Furthermore, the controls did not show alterations in any of the assessed parameters, giving strong support for unchanged soleus fascicle behavior after our intervention period.”

References:

1. Werkhäusen A, Cronin NJ, Albracht K, Paulsen G, Larsen AV, Bojsen-Møller J, Seynnes OR. 2019 Training-induced increase in Achilles tendon stiffness affects tendon strain pattern during running. PeerJ 7.

Comment: Exercise protocol: Given that a single strength training exercise was used in the intervention I would strongly recommend that authors include an image of the exercise apparatus and set-up. I appreciate there are currently a high number of figures included but I would contend this is important for both scientific replication and applied practice.

Response: Thanks for this comment. We added the following figure including a descriptive caption in the revised version of the manuscript. Please note that it has been placed in the supplementary material due to the limited space available.

Figure 1: A conventional leg press was used for the muscle-tendon training of the m. triceps surae. The isometric plantar flexion contractions were performed at 5° dorsiflexion with the knee extended in a seating position (A). The leg press was instrumented with a force sensor in order to control the training stimulus by providing the participant with a visual feedback of the actual contraction intensity. The feedback curve was displayed together with the evidence-based loading profile defined by a sequence of 4 repetitions of 3 s loading and relaxation at 90% of the weekly-adjusted maximum voluntary plantar flexor strength in each of the 5 sets per session, 4 times a week (B).

Comment: Line 147: Why was 2.5 m/s used as the speed for all participants? Was this sufficiently slow enough to ensure a plateau in oxygen consumption and RER value of <1 during the collection period?

Response: A running velocity of 2.5 m/s was used in order to ensure that all participants ran at steady-state, which is a key aspect for the assessment of running economy. The plateau of the oxygen consumption was visually confirmed for each individual and trial and a representative example curve is given in the supplementary material. The average RER for the control group was pre 0.89 ± 0.05 and post 0.87 ± 0.13 and for the intervention group pre 0.94 ± 0.04 and post 0.95 ± 0.05 , respectively. We added to the following sentence to be more clear in the revised manuscript (page: 4, line: 145):

“During an 8-minute running trial on a treadmill at 2.5 m/s, expired gas analysis was conducted and rate of oxygen consumption ($\dot{V}O_2$) and carbon dioxide production ($\dot{V}CO_2$) was calculated as average of the last three minutes [15]. Running economy was then expressed in units of energy [4,30] as Energetic cost = $16.89 \cdot \dot{V}O_2 + 4.84 \cdot \dot{V}CO_2$, where the energetic cost is presented in [W/kg] and $\dot{V}O_2$ and $\dot{V}CO_2$ in [ml/s/kg] [14,15]. Steady state was visually confirmed by the rate of ($\dot{V}O_2$) during each trial and a RER of <1.0 was controlled for during the post analysis (ESM for details).”

Comment: Line 149: The citation here is a paper comparing methods of quantifying energy cost of running. It is not clear which method was used without referring to the supplementary material.

Response: Many apologies. The cited study investigated the appropriateness of the used formula and this was the reason for inserting only this reference. In the revised manuscript, we added the original study and also presented the formula in the main text (see previous comment). Please note that several detailed information is presented in the supplementary material because of the limited space in the main text.

Comment: Line 205: Which post-hoc adjustment was used?

Response: A Benjamini-Hochberg correction was applied and adjusted p-values are reported. This information can be found in the supplementary material (section statistics).

Comment: Line 210: How were the effect sizes interpreted?

Response: Effect sizes were interpreted according to Cohen 1988, where $0.2 \leq g < 0.5$ indicates a small, $0.5 \leq g < 0.8$ indicates a medium, and $g \geq 0.8$ indicates a large effect size. This information can be found in the supplementary material (section statistics).

Comment: Line 229: There appears to be a word missing in this sentence. ‘an altered lengthening-shortening behaviour’ or similar

Response: Thank you for this comment. What we intended to say here is a general description of the MTU behavior that does not refer to intervention effects. Therefore, the behavior is not “altered”. We think that this solves a language issue.

Comment: Line 272: It would be more accurate to discuss the change in economy in the context of within-participant variability (measurement error), rather than between-participant variability, which depends on the sample. A subtle tweak to wording and the reference (eg Blagrove et al., 2017, doi;

10.1080/17461391.2017.1364301; Shaw et al., 2013, doi: 10.1139/apnm-2013-0055) here would provide a more compelling that the 4% improvement is indeed real.

Response: Thank you for this valuable comment. The sentence was changed accordingly in the revised manuscript and one mentioned references were added (page: 7, line: 262):

“Following the intervention, the energetic cost of running was significantly reduced by about 4%, a quantity reported to be above test-retest typical errors [38] and to substantially enhance endurance running performance [39].”

Comment: Line 304-305: Why does the higher maximum plantar flexion moment indicate hypertrophy has occurred? It would be unusual to expect substantial hypertrophy with short-duration isometric contractions. Why can the improvements in strength not be explained as neural adaptation? If so, the discussion below this statement will need to be amended.

Response: We agree with the reviewer that neural adaptation could have contributed to the obtained strength gains following training besides hypertrophy. While there is evidence that neural adaptations may precede morphological responses during the early weeks of strength training onset (1), the intervention duration of our study was quite long at 14 weeks. Several studies have shown an increasing contribution of morphological changes (hypertrophy) following the first 5-6 weeks of training (2,3) beyond neural adaptations (2, 5). Moreover, strength training using explicitly isometric contractions have been shown to provide a sufficient stimulus to induce muscle hypertrophy (6,7).

In our study, EMG_{max} obtained during the maximum voluntary plantar flexions was not changed following the training (pre 0.409 ± 0.114 mV and post 0.410 ± 0.092 mV, $p = 0.300$). Similarly, the training had no effect on the antagonistic co-activation (tibialis anterior EMG 0.034 ± 0.016 mV and post 0.034 ± 0.013 mV, $p = 0.923$). Taken together, the absence of changes in these parameters may not exclude it on other structural levels but strongly indicate that neural aspects may not be the primary course of the found strength gains after the 14 weeks of training.

According to the reviewers comment we changes our formulation in the revised manuscript (page: 8, line: 293):

“However, the higher maximum plantar flexion moment *along with no significant changes in EMG_{max} during the MVCs (pre 0.409 ± 0.114 mV and post 0.410 ± 0.092 mV, $p = 0.300$) and antagonistic co-activation (tibialis anterior EMG 0.034 ± 0.016 mV and post 0.034 ± 0.013 mV, $p = 0.923$) as measures for neural adaption after training strongly indicate muscle hypertrophy, resulting in a 13% increase of F_{max} (pre 2903 ± 750 N, post 3285 ± 831 N).”*

References

1. Folland DJP, Williams AG. 2007 Morphological and Neurological Contributions to Increased Strength. *Sports Med* **37**, 145–168.
2. Narici MV, Hoppeler H, Kayser B, Landoni L, Claassen H, Gavardi C, Conti M, Cerretelli P. 1996 Human quadriceps cross-sectional area, torque and neural activation during 6 months strength training. *Acta Physiologica Scandinavica* **157**, 175–186.
3. Häkkinen K, Komi PV. 1983 Electromyographic changes during strength training and detraining. *Med Sci Sports Exerc* **15**, 455–460.
4. Arampatzis A, Karamanidis K, Albracht K. 2007 Adaptational responses of the human Achilles tendon by modulation of the applied cyclic strain magnitude. *J. Exp. Biol* **210**, 2743–2753.
5. Erskine RM, Jones DA, Williams AG, Stewart CE, Degens H. 2010 Resistance training increases in vivo quadriceps femoris muscle specific tension in young men. *Acta Physiol (Oxf)* **199**, 83–89.
6. Davies J, Parker DF, Rutherford OM, Jones DA. 1988 Changes in strength and cross sectional area of the elbow flexors as a result of isometric strength training. *Eur J Appl Physiol Occup Physiol* **57**, 667–670.
7. Jones DA, Rutherford OM. 1987 Human muscle strength training: the effects of three different regimens and the nature of the resultant changes. *J Physiol* **391**, 1–11.

Comment: Line 346: 'a' seems to be a typographical error here.

Response: 'a' was deleted. Thanks for the hint.

Comment: Line 346: The ref. 16 study (Fletcher et al., 2010) did not find a significant change in running economy following a calf strengthening intervention.

Response: The respective citation was deleted.

Comment: Line 350: There appears to be a word missing between 'training' and 'may'

Response: Thank you for the comment. We corrected this in the revised manuscript.

Comment: Line 355: 'endurance performance' should read 'running economy' here as no performance measures were taken.

Response: We changed the term in the revised manuscript, thank you.

Comment: It has long been recognised that the soleus possesses a high proportion of slow twitch muscle fibres compared to other muscle groups (eg Gollnick et al., 1974, doi: 10.1007/BF00587415). Clearly it is possible to make the soleus stronger and given its role in locomotion and energy cost during exercise, it would certainly make sense for runners to strengthen the muscle. However, do authors think that the soleus has a limited capacity to improve its maximal force output due to its morphological characteristics? The intervention applied here would certainly be novel for the participants, thus beneficial, but would long-term engagement with this type of training for soleus continue to yield benefits in running economy?

Response: Thank you for this important comment. There are reports that fast-twitch fibres feature a greater hypertrophic response to resistance training compared to slow-twitch fibres (1,2,3). Therefore, one might suggest that the soleus muscle is limited in its capacity to improve its maximal force following training. However, the findings are inconsistent and there are studies reporting similar training-induced hypertrophy in slow and fast-twitch fibres (4,5). Therefore, we can argue that the morphological characteristics of the soleus muscle might not be the limiting factor. However, based on the present study we can conclude that two mechanisms contribute to the advantageous work generation by soleus, i.e. the operating enthalpy efficiency and operating force-length potential. The force-length potential was already high throughout the entire stance phase both before and after the training intervention (pre 0.89%, post 0.88%). The enthalpy efficiency throughout the stance was influenced by the intervention and increased by 7% to 92% of the maximum efficiency. Thus, the potential available adaptation range of the enthalpy efficiency for further improvements due to prolonged training seems to be the limiting factor.

References:

1. Hortobágyi T, Hill JP, Houmard JA, Fraser DD, Lambert NJ, Israel RG. 1996 Adaptive responses to muscle lengthening and shortening in humans. *J Appl Physiol* (1985) **80**, 765–772.
2. Andersen JL, Aagaard P. 2000 Myosin heavy chain IIX overshoot in human skeletal muscle. *Muscle Nerve* **23**, 1095–1104.

3. Aagaard P, Andersen JL, Dyhre-Poulsen P, Leffers AM, Wagner A, Magnusson SP, Halkjaer-Kristensen J, Simonsen EB. 2001 A mechanism for increased contractile strength of human pennate muscle in response to strength training: changes in muscle architecture. *J Physiol* **534**, 613–623.
4. Mero AA *et al.* 2013 Resistance training induced increase in muscle fiber size in young and older men. *Eur J Appl Physiol* **113**, 641–650.
5. Bogdanis GC, Tsoukos A, Brown LE, Selima E, Veligeas P, Spengos K, Terzis G. 2018 Muscle Fiber and Performance Changes after Fast Eccentric Complex Training. *Med Sci Sports Exerc* **50**, 729–738.

Referee: 2

General comments:

Comment: In this study, the authors examined the effects of a resistance training program on running economy, and additionally examined how changes in running economy were associated with changes in estimated soleus muscle strength, Achilles tendon stiffness, and operating soleus muscle efficiency, force-length, and force-velocity behaviour. This study provides insight into the mechanisms that may underly improvements in running economy with resistance training. The majority of our understanding of the role of series elasticity on efficiency is from controlled in situ or simulation studies. Thus, this study also provides novel insight into the implications of in vivo muscle and tendon properties during real-world conditions. This manuscript is well-written and interesting to read, and the methods appear sound and appropriate for addressing the research questions. I only have a few comments below that aim to clarify details of the methodology and interpretation of the results.

Response: Thank you for your thorough and valuable comments.

Specific Comments:

Comment: 1. Lines 137-144: Given that increased plantar flexor strength and tendon stiffness are identified as a possible mechanism underlying the main results of this paper, it would be helpful to provide further details of how these variables were measured rather than referring readers to other papers.

For example, in Supplementary material 1, section 2:

“Furthermore, the contribution of the antagonistic muscles to the ankle joint moment was considered by means of an EMG-based method [4].” What specific method was this?

“which was determined using the tendon-excursion method [5,6] and corrected for tendon alignment during the contraction [7].” How were the moment arms corrected for tendon alignment?

“The MTJ displacement artefacts due to an unavoidable change in the ankle joint angle during the MVCs was corrected [8] and the five contractions were averaged to give a reliable measure of the elongation [9]. The AT stiffness was calculated between 50% and 100% of the maximum tendon force using linear regression [10]” How were the changes in ankle joint angle corrected?

Currently the reader would have to consult a range of other papers to fully understand the methods and their justification. More details of these methods and less reliance on previous works would be beneficial.

Response: Thanks for this comment. Please find below the more detailed descriptions of the respective methods that were also included in the revised version of the supplementary material:

- a. Consideration of the contribution of the antagonistic muscles:

The contribution of the antagonistic muscles to the measured ankle joint moments in the different joint angles was considered by a previously reported EMG-based approach [27]. For this reason, the EMG activity of the antagonistic tibialis anterior muscle during the maximum plantar flexions (MVC) was recorded. In separate trials, an individual relationship of EMG amplitude of the tibialis anterior muscle, agonistic moment as well as ankle joint angle was then established. Thereto, the EMG activity of tibialis anterior was measured at rest and during two submaximal isometric dorsal flexion contractions that displayed slightly lower and higher EMG magnitudes as during the maximum plantar flexions [27] in three different joint angles (i.e. dorsi flexion, neutral position and plantar flexion) within the assessed range of motion. The relationship was described by the regression equation $M_{coact} = EMG_{tib. ant.} \cdot (a + b \cdot \alpha_{ankle} + c \cdot \alpha_{ankle}^2)$, where M_{coact} is the antagonistic joint moment during the maximum plantar flexion, $EMG_{tib. ant.}$ is the respective tibialis anterior EMG activity during the MVCs, α_{ankle} the ankle joint angle measured via the Vicon system and a , b and c the individual regression coefficients. Thus, for each joint angle the relationship between moment and EMG activity was assumed to be linear because of the small differences of the EMG magnitude of the two submaximal isometric dorsal flexion contractions [27]. Further, the ankle joint angle-moment relationship presented by the three different measured angles was formulated by a quadratic function to account for the force-length dependence of the muscle. The EMG activity of the tibialis anterior and soleus muscle was measured using a wireless EMG system (Myon m320RX, Myon AG, Baar, Switzerland) and two bipolar surface electrodes (2 cm inter-electrode distance) that were placed on the muscle at an acquisition frequency of 1000 Hz, synchronized with the kinematic data.

b. Tendon excursion method and alignment correction:

The Achilles tendon lever arm was determined for each participant by using the tendon excursion method [3,4]. In this method, the lever arm of the Achilles tendon is calculated as the ratio of the m. gastrocnemius medialis myotendinous junction displacement obtained by ultrasonography at 25 Hz to the corresponding angular excursion of the ankle joint during a passive joint rotation by the dynamometer (5°/s). The ratio was calculated over the interval of 5° dorsiflexion to 10° plantar flexion, where tendon deformation is negligible [8] and five passive rotation trials were averaged to ensure high reliability [29]. The lever arm values were further corrected for the alignment of the tendon occurring during contractions using the factor provided by Maganaris et al. (1998) [5].

c. MTJ displacement artefacts:

The corresponding AT elongation during the ramp MVCs was analyzed based on the displacement of the gastrocnemius medialis-myotendinous junction (MTJ) visualized by B-mode ultrasonography captures (My Lab 60, Esaote, Genova, Italy, 25 Hz). The MTJ displacement artefacts due to an unavoidable increase in the plantar flexion angle during the MVCs were taken into account as they significantly affect the tendon elongation measurement [8]. For this reason, the MTJ displacement as a function of the ankle joint angle was analyzed in an additional trial where the ankle joint was passively rotated by the Biodex over the full range of motion at 5°/s and then used to correct the angle-dependent displacements obtained during the MVCs. The force and elongation data of five ramp MVCs were averaged to give a reliable measure of the AT elongation [9].

Comment: 2. Similar to 1., given that running economy is an important variable in this paper, further details in the main text would be helpful. Since the section “Energetic cost of running” in supplementary material 1 is only one paragraph long, could this not be included in the methods section of the main text? I realize the authors may be limited in terms of length; however, these details are important for interpreting the results of this paper. Similarly, at least the first paragraph of the section “Statistics” in

supplementary material 1 could be included in the main text. Important methods that could affect interpretation of results and conclusions should be easy for readers to access in the main text.

Response: According to the reviewer's comment we added several information of the energetic cost assessment to the main manuscript (page: 4, line: 145, see below) and provided an extended description in the supplementary material. However, we needed to keep the details of the calculation of the required sample size (power analysis) in the supplementary material and only presented the results of this analysis in the main text because of the very limited space available.

"During an 8-minute running trial on a treadmill at 2.5 m/s, expired gas analysis was conducted and rate of oxygen consumption ($\dot{V}O_2$) and carbon dioxide production ($\dot{V}CO_2$) was calculated as average of the last three minutes [15]. Running economy was then expressed in units of energy [4,30] as $\text{Energetic cost} = 16.89 \cdot \dot{V}O_2 + 4.84 \cdot \dot{V}CO_2$, where the energetic cost is presented in [W/kg] and $\dot{V}O_2$ and $\dot{V}CO_2$ in [ml/s/kg] [14,15]. Steady state was visually confirmed by the rate of ($\dot{V}O_2$) during each trial and a RER of <1.0 was controlled for during the post analysis (ESM for details)."

Comment: 3. Line 194: Why did the authors use an efficiency-velocity function rather than a more established metabolic power function (e.g. Minetti & Alexander, 1997 or Umberger, 2010, etc.)? Mechanical work and metabolic cost depend on factors other than just velocity, so why is an efficiency function that depends only on velocity, instead of separately estimating mechanical work and metabolic cost that depend on muscle velocity, length, activation, etc., appropriate for this study? Further explanation/justification in the text would be helpful.

Response: Thank you for this important comment. In our opinion, the accurate assessment of muscular work in vivo is currently an unresolved problem because the required muscle force for the calculation cannot be measured. Of course, there are several studies in the literature that estimate muscle forces and muscular work by means of inverse dynamics approaches and musculoskeletal models. With all the respect towards these studies, we believe that the assumptions taken in such approaches are very strong and may dramatically affect the calculated muscular work values. Taking into consideration the relevant methodological limitations, we decided not to include calculations of force/mechanical work in our study. Rather, we tried to develop a methodological design that allows us to consider experimentally-assessed basic mechanisms for muscle force and work production (i.e. muscle force potential due to the force-length and force-velocity relationship, muscle activity and enthalpy efficiency-velocity relationship) and to investigate these mechanisms during running. Muscle length and activation can affect the heat rate, most likely due to actomyosin interaction and sarcoplasmic reticular ion transport (1), which can in turn influence the enthalpy efficiency-velocity relationship and thus might be considered as additional scale factors in our approach. However, we found a continuous shortening of the soleus fascicles very close to the optimal length and mainly in the ascending part of the force-length curve. Heat rate is nearly maximal at the optimum muscle length and there are only small changes in lengths shorter than the optimum (1,2). Furthermore, taken into consideration that the soleus fascicles operated at the same length in the pre and post condition (similar force-length potential) and the activated muscle volume based on our calculations using the Hill-type model did not show relevant pre-post differences, we can argue that both muscle operating length and muscle activation did not affect our outcomes regarding the enthalpy-efficiency.

References:

1. Woledge RC, Curtin NA, Homsher E. 1985 Energetic aspects of muscle contraction. *Monogr Physiol Soc* **41**, 1–357.
2. Hilber K, Sun Y-B, Irving M. 2001 Effects of sarcomere length and temperature on the rate of ATP utilisation by rabbit psoas muscle fibres. *The Journal of Physiology* **531**, 771–780.

Comment: Also, the fitted values in Table 1 of Hill (1967) are for frog muscle at 0 degrees C. Since frogs are ectotherms, the muscle temperature would be near that of the external environment, far below physiological temperature for human muscle. This could affect both muscle force and velocity (see James, 2013 for review) and therefore the fitted function. Additionally, amphibian muscle contains larger concentrations of parvalbumin compared to terrestrial muscles, which can alter the heat rate and estimated metabolic cost (Woledge et al., 1985, pp. 257-260). What are the implications of these considerations on the results of this study?

James, R. S. (2013). A review of the thermal sensitivity of the mechanics of vertebrate skeletal muscle. *Journal of Comparative Physiology B*, 183(6), 723-733.

Woledge, R. C., Curtin, N. A., & Homsher, E. (1985). Energetic aspects of muscle contraction. *Monographs of the Physiological Society*.

Response: We agree with the reviewer that there is evidence for an effect of temperature on efficiency measures in both amphibian and mammalian muscles (1,2,3). Thus, it is possible that the maximum enthalpy (mechanical) efficiency of 0.44 of the frog muscle from the Hill (1964) paper (4), that we used for our analysis, is higher under more physiological temperatures. As a reference for the human soleus muscle, a maximum efficiency value of 0.48 could be taken from the murine soleus muscle under almost physiological conditions (30°C) and comparable fiber type composition (2). Please note that this value of the maximum efficiency is close to the value reported for the frog muscle (0.44) by Hill (1964) (4). Besides, since we calculated efficiency as a function of the soleus muscle shortening velocity (adjusted for physiological temperature) and only discussed our findings in terms of percentage change of the enthalpy efficiency, any discrepancies regarding the magnitude of the enthalpy efficiency would not significantly affect our results.

Methodologically more important for our results would be a significant difference of the shape of the efficiency-velocity curve with great shifts of the velocity at maximum efficiency. Again the study by Barclay et al. (2010) (2) on the soleus mouse muscle showed that temperature had no effect on the velocity on the maximum efficiency in the investigated range of 20 to 30°C (between 0.19 and 0.20 V/V_{max}) and shape. The reported velocity for the maximum efficiency value at 30° for the mouse soleus muscle (0.19 V/V_{max} , table 1 in (2)) was very close to the value of the frog muscle provided by the paper of Hill (0.18 V/V_{max}), which suggests similarity between efficiency-velocity curves in further support of our analysis.

Moreover, in our reported sensitivity analysis we tried to examine the effect of changes in the shape of the curve and changes of V_{max} by a) changing V_{max} in 10%-intervals and b) replacing the curve from the frog muscle of the Hill paper (4) by the data presented by Barclay et al. (1993) (5) for the soleus mouse muscle. The findings showed that the significant pre to post enthalpy efficiency increase for the MTU lengthening phase and entire stance phase persisted for values between $V_{max-30\%}$ and $V_{max+10\%}$ both using the data of Hill or Barclay et al. ($p < 0.05$), which confirms and strengthens the observed intervention effect (detailed descriptive values and p-values see suppl. material 2).

The following changes were made in the revised limitation section to be more clear (starting page: 9, line: 358):

“[...] We evaluated the effect of a) decreasing V_{max} by 10% intervals and b) replacing the underlying enthalpy efficiency values measured at the frog sartorius at 0°C from Hill (1964) [20] by the data presented by Barclay et al. (1993) [22] for the predominantly slow fiber type soleus mouse muscle at 21°C, comparable to the human soleus muscle.

[...] Furthermore, since we calculated the efficiency as a function of the soleus muscle shortening velocity (adjusted for physiological temperature) and only discussed our findings in terms of percentage change, any uncertainties about the magnitude of the enthalpy efficiency would not affect our results.”

References:

1. James RS. 2013 A review of the thermal sensitivity of the mechanics of vertebrate skeletal muscle. *J Comp Physiol B* **183**, 723–733.
2. Barclay CJ, Woledge RC, Curtin NA. 2010 Is the efficiency of mammalian (mouse) skeletal muscle temperature dependent? *The Journal of Physiology* **588**, 3819–3831.
3. He ZH, Bottinelli R, Pellegrino MA, Ferenczi MA, Reggiani C. 2000 ATP consumption and efficiency of human single muscle fibers with different myosin isoform composition. *Biophys J* **79**, 945–961.
4. Hill AV. 1964 The efficiency of mechanical power development during muscular shortening and its relation to load. *Proceedings of the Royal Society of London. Series B. Biological Sciences* **159**, 319–324.
5. Barclay CJ, Constable JK, Gibbs CL. 1993 Energetics of fast- and slow-twitch muscles of the mouse. *The Journal of Physiology* **472**, 61–80.
6. Barclay CJ. 2015 Energetics of contraction. *Compr Physiol* **5**, 961–995.
7. Nelson FE, Ortega JD, Jubrias SA, Conley KE, Kushmerick MJ. 2011 High efficiency in human muscle: an anomaly and an opportunity? *J Exp Biol* **214**, 2649–2653.

Comment: 4. Line 268: "... the results provide additional evidence that a combination of greater plantar flexor muscle strength and Achilles tendon stiffness decrease the energy cost of running [14,15] and indicate that the soleus enthalpy efficiency is a contributive determinant." It's alluded to with "a combination" but consider an additional sentence here noting that an increase in stiffness by itself may not increase efficiency. Later in line 349 the authors state "strength increases without concomitant stiffening of the AT after a period of training may increase levels of operating and maximum AT strain [24], which have been associated with pathologies [53] but also possible functional decline [54]." Function may also decline with increases in stiffness without concomitant increases in muscle strength. For example, see Figure 5 in Lichtwark and Wilson (2005) in which muscle efficiency during running decreased with increases in AT stiffness beyond the optimal stiffness.

Lichtwark, G. A., & Wilson, A. M. (2007). Is Achilles tendon compliance optimised for maximum muscle efficiency during locomotion? *Journal of Biomechanics*, 40(8), 1768-1775

Response: In agreement with the reviewer, we tried to make clear that it is in fact the combination of a higher muscle strength and increased tendon stiffness that potentially improved the efficiency of the operating soleus muscle and not tendon stiffness or muscle strength alone. The found increases in muscle strength and tendon stiffness were always reported alongside each other throughout the entire manuscript. Such a "balanced" adaptation seem required to facilitate the functional interplay of muscle and tendon in a way that tendon compliance can be optimally used for movement performance and efficiency but also tendon health (1,2,3,4). According to the reviewer's comment we now added a sentence in the respective section as follows (page: 9, line: 343):

"Strength increases without concomitant stiffening of the AT after a period of training can increase levels of operating and maximum AT strain [24], which have been associated with pathologies [53] but also possible functional decline [54]. On the other hand, increased stiffness without higher muscle strength may limit function by reducing relevant operating tendon strains as well (2). In our study, the maximum AT strain during the MVCs was not affected by the ..."

References:

1. Arampatzis A, Mersmann F, Bohm S. 2020 Individualized Muscle-Tendon Assessment and Training. *Front. Physiol.* 11.
2. Lichtwark GA, Wilson AM. 2007 Is Achilles tendon compliance optimised for maximum muscle efficiency during locomotion? *J Biomech* 40, 1768–1775.
3. Orselli MIV, Franz JR, Thelen DG. 2017 The effects of Achilles tendon compliance on triceps surae mechanics and energetics in walking. *J Biomech* 60, 227–231.
4. Uchida TK, Hicks JL, Dembia CL, Delp SL. 2016 Stretching Your Energetic Budget: How Tendon Compliance Affects the Metabolic Cost of Running. *PLoS ONE* 11, e0150378.

Comment: 5. Line 291: “The exercise-induced increase in muscle strength and AT stiffness resulted in an alteration of the operating fascicle velocity profile that led to a significant increase of the enthalpy efficiency of the operating soleus [...], improving the efficiency of muscular work production.” Because the only factor that was manipulated in this study was the exercise intervention, changes in muscle strength, AT stiffness, fascicle velocities, and enthalpy efficiency are only associated with one another rather than there being any causal relationship between them.

Response: Thanks for this comment. We softened our formulation accordingly (page 8:, line: 281):

“The exercise-induced increase in muscle strength and AT stiffness was associated with an alteration of the operating fascicle velocity profile and a significant increase of the enthalpy efficiency of the operating soleus in the phase of MTU lengthening (88% of the maximum efficiency), potentially improving the efficiency of muscular work production.”